# From Finite to Countable-Armed Bandits

**Anand Kalvit[1] and Assaf Zeevi[2]**
Graduate School of Business
Columbia University
New York, USA
{[1]akalvit22,[2]assaf}@gsb.columbia.edu

## Abstract

We consider a stochastic bandit problem with countably many arms that belong to a finite set of types, each characterized by a unique mean reward. In addition, there is a fixed distribution over types which sets the proportion of each type in the population of arms. The decision maker is oblivious to the type of any arm and to the aforementioned distribution over types, but perfectly knows the total number of types occurring in the population of arms. We propose a fully adaptive online learning algorithm that achieves $\mathcal{O}\left(\log n\right)$ distribution-dependent expected cumulative regret after any number of plays $n$, and show that this order of regret is best possible. The analysis of our algorithm relies on newly discovered concentration and convergence properties of optimism-based policies like UCB in finite-armed bandit problems with *zero gap*, which may be of independent interest.

## 1 Introduction

**Background and motivation.** The multi-armed bandit (MAB) problem is a widely studied machine learning paradigm that captures the tension between *exploration* and *exploitation* in online decision making. The problem traces its roots to 1933 when it was first studied in the context of clinical trials in [21]. It has since evolved and numerous variants of the MAB problem have seen an upsurge in applications across a plethora of domains spanning dynamic pricing, online auctions, packet routing, scheduling, e-commerce and matching markets to name a few (see [12] for a comprehensive survey). In its simplest formulation, the decision maker must sequentially play an arm at each time instant out of a set of $K$ possible arms, each characterized by its own distribution of rewards. The objective is to maximize cumulative expected payoffs over the horizon of play. Every play of an arm results in an independent sample from its reward distribution. The decision maker, oblivious to the statistical properties of the arms, must balance exploring new arms and exploiting the best arm played thus far. The objective of maximizing cumulative rewards is often converted to minimizing *regret* relative to an oracle with perfect ex ante knowledge of the best arm. The seminal work [20] was the first to show that the optimal order of this regret is asymptotically logarithmic in the number of plays. Much of the focus since has been on the design and analysis of algorithms that can achieve near-optimal regret rates (see [5, 16, 15], etc., and references therein).

Many practical applications of the multi-armed bandit problem involve a prohibitively large number of arms, the number in some cases is even larger than the horizon of play itself. This renders finite-armed models unsuitable vehicle for the study of such settings. The simplest prototypical example of such a setting occurs in the context of online assignment problems arising in large marketplaces serving a very large population of agents that each belong to one of $K$ possible types; e.g., if $K = 2$, the set of agent types could be {"high caliber", "low caliber"}, {"patient", "impatient"}, etc. Such finite-typed settings are also relevant in many applications with an exponentially large choice space and where a limited planning horizon forbids exploration-exploitation in the traditional sense (This is common in online retail where assortments of substitutable products are selected from a very

large product space, cf. [2]). We shall refer to problems of this nature as *countable-armed bandits (CAB)*. The CAB problem lies hedged between the finite-armed bandit problem on one end, and the so called *infinite-armed bandit problem* on the other. As the name suggests, the latter is typically characterized by a continuum of arm types and for this reason, the CAB problem is closer in spirit to the finite-armed problem despite an infinity of arms, though it has its own unique salient features.

The CAB problem is characterized by a finite set of arm types $\mathcal{T}$ and a distribution over $\mathcal{T}$ denoted by $\mathcal{D}(\mathcal{T})$. Since this is the first systematic investigation of said bandit model, we assume in this paper that $|\mathcal{T}| = 2$ for a clear exposition of key technical results and proof ideas unique to the countable-armed setting. The statistical complexity of the CAB problem with a binary $\mathcal{T}$ is determined by three primitives: (i) the sub-optimality gap ($\Delta$) between the mean rewards of the superior and inferior arm types; (ii) the proportion of arms of the superior type in the infinite population of arms ($\alpha$); and (iii) the duration of play ($n$).

**Main contributions.** We show that the finite-time expected cumulative regret achievable in the CAB problem, absent ex ante knowledge of $(\Delta, \alpha, n)$, is $\mathcal{O}\left(\beta_\Delta^{-1}\left(\Delta^{-1}\log n + \alpha^{-1}\Delta\right)\right)$ (Theorem 3), where $\beta_\Delta \leqslant 1$ is an instance-specific constant that depends only on the reward distributions associated with the arm types, and the big-Oh notation only hides absolute constants. To this end, we propose a fully adaptive online learning algorithm that has the aforementioned regret guarantee and show that its performance cannot essentially be improved upon. The proof of Theorem 3 relies on a newly discovered concentration property of optimism-based algorithms such as UCB in finite-armed bandit problems with *zero gap*, e.g., a two-armed bandit with $\Delta = 0$ (Theorem 4 (i)). This result is of independent interest as it disproves a folk conjecture on non-convergence of UCB in zero gap settings (Theorem 4 (ii)) and is likely to have implications for statistical inference problems involving adaptive data collected by UCB-like algorithms. Additionally, the zero gap setting also highlights a stark difference between the limiting pathwise behavior of UCB and Thompson Sampling. In particular, we observe empirically that UCB's concentration and convergence properties à la Theorem 4 are, in fact, violated by Thompson Sampling (Figure 2). A theoretical explanation for said pathological behavior of Thompson Sampling is presently lacking in literature. Before describing the CAB model formally, we survey two closely related MAB models below and note key differences with our model.

**Relation to the finite-armed bandit model.** In this problem, finiteness of the action set (set of arms) allows for sufficient exploration of all the arms which makes it possible to design policies that achieve near-optimal regret rates (cf. [5, 15], etc.) relative to the lower bound in [20]. In contrast, exploring every single arm in our problem is: (a) infeasible due to an infinity of available arms; and (b) clearly sub-optimal since any attempt at it would result in linear regret. The fundamental difficulty in the countable-armed problem lies in identifying a consideration set that contains at least one arm of the optimal type. In the absence of any ex ante information on $(\Delta, \alpha)$, it is unclear whether this can be done in a manner that would guarantee sub-linear regret; and secondly, what is the minimal achievable regret. These questions capture the essence of our work in this paper.

**Relation to the infinite-armed bandit model.** This problem also considers an infinite population of arms and a fixed *reservoir* distribution over the set of arm types, which maps to the set of possible mean rewards. However, unlike our problem, the set of arm types here forms the continuum $[0, 1]$. The infinite-armed problem traces its roots to [7] where it was first studied under a Bernoulli reward setting with the reservoir distribution of mean rewards being Uniform on $[0, 1]$. This work spawned a rich literature on infinite-armed problems, however, to the best of our knowledge, all of the extant body of work is predicated on the assumption that the reservoir distribution satisfies a certain regularity property (or a variant thereof) in the neighborhood of the optimal mean reward (cf. [7, 22, 9, 13, 11] for a comprehensive survey). Such assumptions restrict the set of types to infinite cardinality sets. In terms of statistical complexity, this has the implication that the minimal achievable regret is polynomial in the number of plays. In contrast, the CAB model is fundamentally simpler since the set of arm types is only finite. The natural question then is if better regret rates are possible for the CAB problem at least on "well-separated" instances. This is the central question underlying our work.

In addition to the infinite-armed bandit model discussed above, there are two other related problem classes: *continuum-armed bandits* and *online stochastic optimization*. However, these problems are predicated on an entirely different set of assumptions involving the topological embedding of the arms and regularities of the mean-reward function, and share little similarity with our stochastic model. The reader is advised to refer to [17, 1, 19, 6, 18, 10], etc., for a detailed coverage of the aforementioned problem classes.

**Organization of the paper.** The CAB problem is formally described in § 2. Algorithms for the CAB problem and related theoretical guarantees are stated in § 3. A formal statement of the concentration and convergence properties of UCB in finite-armed bandits with zero gap is deferred to § 4. Proof sketches are included in the main text to the extent permissible, full proofs and other technical details including ancillary lemmas are relegated to the appendices.

## 2   Problem formulation

The set of arm types is denoted by $\mathcal{T} = \{1, 2\}$. Each type $i \in \mathcal{T}$ is characterized by a *unique* mean reward $\mu_i \in (0, 1)$ with the rewards themselves bounded in $[0, 1]$. The proportion of arms of type $\arg\max_{i \in \mathcal{T}} \mu_i$ in the population of arms is given by $\alpha$. Different arms of the same type may have distinct reward distributions but their mean rewards are equal. For each $i \in \mathcal{T}$, $\mathcal{G}(\mu_i)$ denotes a finite[1] collection of reward distributions with mean $\mu_i$ associated with the type $i$ sub-population.

**Assumption 1 (Maximally supported rewards in $[0, 1]$)** *Any CDF $F \in \bigcup_{i \in \mathcal{T}} \mathcal{G}(\mu_i)$ satisfies: (i)* $\sup\{x \in \mathbb{R} : F(x) = 0\} = 0$, *and (ii)* $\inf\{x \in \mathbb{R} : F(x) = 1\} = 1$.[2]

For example, distributions such as Bernoulli$(\cdot)$, Beta$(\cdot, \cdot)$, Uniform on $[0, 1]$, etc., satisfy Assumption 1. Without loss of generality, we assume $\mu_1 > \mu_2$ and call type 1, the optimal type. $\Delta := \mu_1 - \mu_2$ denotes the separation (or gap) between the types. The index set $\mathcal{I}_n$ contains labels of all the arms that have been played up to and including time $n$ (with $\mathcal{I}_0 := \phi$). The set of available actions at time $n$ is given by $\mathcal{A}_n = \mathcal{I}_{n-1} \cup \{\text{new}\}$ and $\mathcal{P}(\mathcal{A}_n)$ denotes the probability simplex on $\mathcal{A}_n$. At any time $n$, the decision maker must either choose to play an arm from $\mathcal{I}_{n-1}$, or select the action "new" which corresponds to playing a new arm, unexplored hitherto, whose type is an unobserved, independent sample from an unknown distribution on $\mathcal{T}$ denoted by $\mathcal{D}(\mathcal{T}) = (\alpha, 1 - \alpha)$. The realized rewards are independent across arms and i.i.d. in time keeping the arm fixed. The natural filtration $\mathcal{F}_n$ is defined w.r.t. the sequence of rewards realized up to and including time $n$ (with $\mathcal{F}_0 := \phi$). A policy $\pi = \{\pi_n : n \in \mathbb{N}\}$ is a non-anticipatory adaptive sequence that for each $n$ prescribes an action from $\mathcal{P}(\mathcal{A}_n)$, i.e., $\pi_n : \mathcal{F}_{n-1} \to \mathcal{P}(\mathcal{A}_n) \; \forall \, n \in \mathbb{N}$. The cumulative pseudo-regret of $\pi$ after $n$ plays is given by $R_n^\pi = \sum_{m=1}^n (\mu_1 - \mu_{t(\pi_m)})$, where $t(\pi_m)$ denotes the type of the arm played by $\pi$ at time $m$. We are interested in the problem $\min_{\pi \in \Pi} \mathbb{E} R_n^\pi$, where $n$ is the horizon of play, $\Pi$ is the set of all non-anticipation policies, and the expectation is w.r.t. the randomness in $\pi$ as well as $\mathcal{D}(\mathcal{T})$. We remark that $\mathbb{E} R_n^\pi$ is the same as the traditional notion of expected cumulative regret in our problem[3].

**Other notation.** We reemphasize that for any given arm, *label* and *type* are two distinct attributes. The number of plays up to and including time $n$ of arm $i$ is denoted by $N_i(n)$, and its type by $t(i) \in \mathcal{T}$. At any time $n^+$, $(X_{i,j})_{j=1}^m$ denotes the sequence of rewards realized from the first $m \leqslant N_i(n)$ plays of arm $i$. The natural filtration at time $n^+$ is formally defined as $\mathcal{F}_n := \sigma\left\{(X_{i,j})_{j=1}^{N_i(n)} ; i \in \mathcal{I}_n\right\}$. The empirical mean reward from the first $N_i(n)$ plays of arm $i$ is denoted by $\overline{X}_i(n)$. An absolute constant is understood to be one that does not depend on any problem primitive or free parameters.

## 3   Main results: Rate-optimal algorithms for the CAB problem

In the finite-armed bandit problem, the gap $\Delta$ is the key primitive that determines the statistical complexity of regret minimization. The literature on finite-armed bandits roughly bifurcates into two broad strands of algorithms, $\Delta$-*aware* and $\Delta$-*agnostic*. Explore-then-Commit (aka, Explore-then-Exploit) and $\epsilon_n$-Greedy are two prototypical examples of the former category, while UCB and Thompson Sampling belong to the latter. In the CAB problem too, $\Delta$ plays a key role in determining the complexity of regret minimization. Since this is the first theoretical treatment of the subject matter, it is instructive to first study the $\Delta$-aware case to gain insight into the basic premise that sets the finite and countable-armed problems apart. We investigate the case of a $\Delta$-aware decision maker in § 3.1 and the $\Delta$-agnostic case in § 3.2. Before proceeding to the algorithms, we first state a lower

bound for the CAB problem that applies for any admissible policy. In what follows, an *instance* of the CAB problem refers to the tuple $(\mathcal{G}(\mu_1), \mathcal{G}(\mu_2))$ with $|\mu_1 - \mu_2| = \Delta$, and we slightly overload the notation for expected cumulative regret to emphasize its instance-dependence.

**Theorem 1 (Lower bound on achievable performance)** *For any $\Delta > 0$, $\exists$ a pair of reward distributions $(Q_1, Q_2)$ with means $(\mu_1, \mu_2)$ respectively, satisfying $|\mu_1 - \mu_2| = \Delta$, and an absolute constant $C$, s.t. the expected cumulative regret of any asymptotically consistent[4] policy $\pi$ on the CAB instance $\nu = (\{Q_1\}, \{Q_2\})$ satisfies for all $\alpha \leqslant 1/2$ and $n$ large enough, $\mathbb{E}R_n^\pi(\nu) \geqslant C\Delta^{-1}\log n$.*

**Remark.** Theorem 1 bears resemblance to the classical lower bound of Lai and Robbins for finite-armed bandits [20], but the two results differ in a fundamental way. While $\nu = (\{Q_1\}, \{Q_2\})$ fully specifies a two-armed bandit problem, it is the *realization* of $\nu$, i.e., an infinite sequence $(r_i)_{i \in \mathbb{N}}$ with $\mathbb{P}\left(r_i = Q_{\arg\max_{j \in \{1,2\}} \mu_j}\right) = \alpha$ and where $r_i \in \{Q_1, Q_2\}$ indicates the reward distribution of arm $i \in \mathbb{N}$, that specifies the CAB problem. As such, traditional lower bound proofs for finite-armed bandits are not directly adaptable to the CAB problem. Nonetheless, the two results retain structural similarities because the CAB problem, despite its additional complexity, remains amenable to a standard reduction to a hypothesis testing problem. It must be noted that any policy incurs linear regret when $\alpha = 0$, while zero regret when $\alpha = 1$. Theorem 1 states a uniform lower bound independent of $\alpha$ that applies for all $\alpha \leqslant 1/2$. Since the CAB problem with $\alpha < 1/2$ is statistically harder than its two-armed counterpart, we believe the lower bound in Theorem 1 is in fact, unachievable in the sense of the exact scaling of the $\log n$ term. However, our objective in this paper is to develop algorithms for the CAB problem that are order-optimal in $n$ and to that end, Theorem 1 serves its stipulated purpose. Characterizing an *achievable* scaling of the lower bound and its dependence on $\alpha \in [0, 1]$ remains an open problem. We consider the restriction to the classical asymptotically consistent policy class (Definition 1, Appendix A) as more generic policy classes are unwieldy for lower bound proofs due to reasons stemming from the combinatorial nature of our problem. Full proof is given in Appendix A.

### 3.1 A near-optimal $\Delta$-aware algorithm for the CAB problem

The intuition and understanding developed through this section shall be useful while studying the $\Delta$-agnostic case later and highlights key statistical features of the CAB problem. Below, we present a simple fixed-design ETC (Explore-then-Commit) algorithm assuming ex ante knowledge of the duration of play[5] $n$ and a separability parameter $\delta \in (0, \Delta)$. In what follows, we use *select* to indicate an arm selection action, and *play* to indicate the action of pulling a selected arm. A reward is only realized after an arm is played, not merely selected. A *new* arm refers to one that has never been selected before. $(X_{i,j})_{j=1}^m$ denotes the sequence of rewards realized from the first $m$ plays of arm $i$.

---

**Algorithm 1** ETC-$\infty$(2): ETC for an infinite population of arms with $|\mathcal{T}| = 2$.

---

1: **Input:** $(n, \delta)$, where $\delta \in (0, \Delta]$.
2: Set $L = \lceil 2\delta^{-2}\log n \rceil$. Set budget $T = n$.
3: **Initialization** (Starts a new epoch)**:** Select two *new* arms. Call it consideration set $\mathcal{A} = \{1, 2\}$.
4: $m \leftarrow \min(L, T/2)$.
5: Play each arm in $\mathcal{A}$ $m$ times. Update budget: $T \leftarrow T - 2m$.
6: **if** $\left|\sum_{j=1}^m (X_{1,j} - X_{2,j})\right| < \delta m$ **then**
7: $\quad$ Permanently discard $\mathcal{A}$ and go to **Initialization**.
8: **else**
9: $\quad$ Commit the remaining budget of play to arm $i^* \in \arg\max_{i \in \mathcal{A}} \sum_{j=1}^m X_{i,j}$.

---

**Mechanics of ETC-$\infty$(2).** The horizon of play is divided into epochs of length $2m = \mathcal{O}(\log n)$ each. The algorithm starts off by selecting a pair of arms at random from the infinite population of arms and playing them $m$ times each in the first epoch. Thereafter, the pair is classified as having either identical or distinct types via a hypothesis test through step 6. If classified as "identical," the algorithm permanently discards both the arms (never to be selected again) and replaces them with yet another newly selected pair, which is subsequently played equally in the next epoch. This process is

repeated until a pair of arms with distinct types is identified. In the event of such a discovery, the algorithm commits the residual budget to the empirically better arm in the current consideration set.

**Theorem 2 (Upper bound on the expected regret of ETC-∞(2))** *The expected cumulative regret of the policy $\pi$ given by Algorithm 1 after $n$ plays is bounded as follows:*

$$\mathbb{E}R_n^\pi \leqslant \min\left(\Delta n,\ \Delta\left(2 + \alpha^{-1}\right)\left(2\delta^{-2}\log n + 1\right) + \alpha^{-1}\left(f(n,\delta,\Delta) + 2\right)\Delta\right),$$

*where $f(n,\delta,\Delta) = o(1)$ in $n$ and independent of $\alpha$ (Note: This result is agnostic to Assumption 1.).*

**Proof sketch of Theorem 2.** On a pair of arms of the optimal type (type 1), any playing rule incurs zero regret in expectation, whereas the expected regret is linear in the number of plays if the pair is of the inferior type (type 2). Since it is statistically impossible to distinguish between a type 1 pair and a type 2 pair in the absence of any distributional knowledge of the associated rewards, the algorithm must identify a pair of distinct types whenever so obtained, to avoid high regret. This is precisely done through step 6 of Algorithm 1 via a hypothesis test. Since the distribution over the types, denoted by $\mathcal{D}(\mathcal{T}) = (\alpha, 1 - \alpha)$, is stationary, the number of fresh draws of consideration sets until one with arms of distinct types is obtained is a geometric random variable (say $W$). Thus, it only takes $(\mathbb{E}W)(2m) = \mathcal{O}(\log n)$ plays in expectation to obtain such a pair and identify it correctly with high probability. The algorithm subsequently commits to the optimal arm in the pair with high probability. Therefore, the overall expected regret is also $\mathcal{O}(\log n)$. Full proof is relegated to Appendix B. □

**Remark.** The key idea used in Algorithm 1 is that of interleaving hypothesis testing (step 6) with regret minimization (step 9). In the stated version of the algorithm, the regret minimization step simply commits to the arm with the higher empirical mean reward. The framework of Algorithm 1 also allows for other regret minimizing playing rules (for e.g., $\epsilon_n$-Greedy [5], etc.) to be used instead in step 9. The flexibility afforded by this framework shall become apparent in § 3.2.

## 3.2 A near-optimal $\Delta$-agnostic algorithm for the CAB problem

Designing an adaptive, $\Delta$-agnostic algorithm and the proof that it can achieve the lower bound in Theorem 1 (in $n$, modulo multiplicative constants) is the main focus of this paper. Recall that ex ante information about $\Delta$ serves a dual role in Algorithm 1: (i) in calibrating the epoch length in step 2; and (ii) determining the separation threshold for hypothesis testing in step 6. In the absence of information on $\Delta$, it is a priori unclear if there exists an algorithm that would guarantee sublinear regret on "well-separated" instances. In Algorithm 2 below, we present a generic framework called ALG($\Xi, \Theta, 2$), around which various $\Delta$-agnostic playing rules such as UCB, Thompson Sampling, etc., can be tested. In what follows, $s \in \{1, 2, ...\}$ indicates a discrete time index at which an arm may be played in the current epoch. Every epoch starts from $s = 1$.

---

**Algorithm 2** ALG($\Xi, \Theta, 2$): An algorithmic framework for countable-armed bandits with $|\mathcal{T}| = 2$.

1: **Input:** A $\Delta$-agnostic playing rule $\Xi$, a deterministic sequence $\Theta \equiv \{\theta_m : m = 1, 2, ...\}$ in $\mathbb{R}$.
2: **Initialization** (Starts a new epoch)**:** Select two *new* arms. Call it consideration set $\mathcal{A} = \{1, 2\}$.
3: For $s \in \{1, 2\}$, play each arm in $\mathcal{A}$ once.
4: $m \leftarrow 1$.
5: **for** $s \in \{3, 4, ...\}$ **do**
6:     **if** $\left|\sum_{j=1}^{m}(X_{1,j} - X_{2,j})\right| < \theta_m$ **then**
7:         Permanently discard $\mathcal{A}$ and go to **Initialization**.
8:     **else**
9:         Play an arm from $\mathcal{A}$ according to $\Xi$.
10:         $m \leftarrow \min_{i \in \mathcal{A}} N_i(s)$.

---

**On the issue of sample-adaptivity in hypothesis-testing.** The foremost noticeable aspect of Algorithm 2 that also sets it apart from Algorithm 1, is that the samples used for hypothesis testing in step 6 are collected *adaptively* by $\Xi$. For instance, if $\Xi = $ UCB1 [5], then step 9 translates to playing arm $i^* \in \arg\max_{i \in \mathcal{A}}\left(\overline{X}_i(s-1) + \sqrt{2\log(s-1)/N_i(s-1)}\right)$. This is distinct from the classical hypothesis testing setup used in step 6 of Algorithm 1, where the collected data does not exhibit such dependencies. It is well understood that adaptivity in the sampling process can lead to biased

inferences (see, e.g., [14]). However, for standard choices of $\Xi$ such as UCB or Thompson Sampling (or variants thereof), the exploratory nature of $\Xi$ ensures that the test statistic $\sum_{j=1}^{m}(X_{1,j} - X_{2,j})$ where $m = \min_{i \in \mathcal{A}} N_i(s)$, remains agnostic to any sample-adaptivity due to $\Xi$. This statement is formalized and further explained in Lemma 1 (Appendix F).

**Mechanics of ALG**$(\Xi, \Theta, 2)$**.** We call a consideration set $\mathcal{A}$ of arms "heterogeneous" if it contains arms of distinct types, and "homogeneous" otherwise. Algorithm 2 has a master-slave framework in which step 6 is the master routine and $\Xi$ serves as the slave subroutine in step 9. The purpose of step 6 is to quickly determine if $\mathcal{A}$ is homogeneous, in which case it discards $\mathcal{A}$ and restarts the algorithm afresh in a new epoch. On the other hand, whenever a heterogeneous $\mathcal{A}$ gets selected, step 6 ensures that its selection persists in expectation which allows $\Xi$ to run "uninterrupted." This idea is formalized in Lemma 2 (Appendix F). In a nutshell, Algorithm 2 runs in epochs of random lengths that are themselves determined adaptively. At the beginning of every epoch, the algorithm selects a new consideration set $\mathcal{A}$ and deploys $\Xi$ on it. It then determines (via the hypothesis test in step 6) whether to keep playing $\Xi$ on $\mathcal{A}$ or to stop and terminate the epoch, based on the current sample history of $\mathcal{A}$. Upon termination, $\mathcal{A}$ is discarded and the algorithm starts afresh in a new epoch.

**Calibrating $\Theta$.** ALG$(\Xi, \Theta, 2)$ identifies homogeneous $\mathcal{A}$'s by means of a hypothesis test through step 6. It starts with the null hypothesis $\mathcal{H}_0$ that the current $\mathcal{A}$ is heterogeneous and persists with it until "enough" evidence to the contrary is gathered. If $\mathcal{H}_0$ were indeed true, the Strong Law of Large Numbers (SLLN) would dictate that $\left|\sum_{j=1}^{m}(X_{1,j} - X_{2,j})\right| \sim \Delta m$, almost surely. If $\mathcal{H}_0$ were false, it would follow from the Central Limit Theorem (CLT) that $\left|\sum_{j=1}^{m}(X_{1,j} - X_{2,j})\right| = \mathcal{O}\left(\sqrt{m}\right)$. Therefore, in order to separate $\mathcal{H}_0$ from its complement, the right $\theta_m$ must satisfy: $\theta_m = o(\Delta m)$ and $\theta_m = \omega\left(\sqrt{m}\right)$. Indeed, our choice of $\theta_m$ (see (2)) satisfies these conditions and is such that $\theta_m \sim 2\sqrt{m \log m}$. We reemphasize that the calibration of $\Theta$ is independent of $\Delta$ and only *informed* by classical results (SLLN, CLT) that are themselves inapplicable since the data collection is adaptive.

**High-level overview of results.** We show that for a suitably calibrated input sequence $\Theta$ (see (2)), the instance-dependent expected cumulative regret of ALG(UCB1, $\Theta$, 2) is logarithmic in the number of plays anytime, this order of regret being best possible. We also demonstrate empirically that a key concentration property of UCB1 that is pivotal to the aforementioned regret guarantee, is violated for Thompson Sampling (TS) and therefore, ALG(TS, $\Theta$, 2) suffers linear regret. A formal statement of said concentration property of UCB1 is deferred to § 4. The regret upper bound of ALG(UCB1, $\Theta$, 2) is stated next in Theorem 3. Following is an auxiliary proposition that is useful towards Theorem 3.

**Proposition 1 (Lower bound on the true negative rate)** *For each $i \in \mathcal{T} = \{1, 2\}$, let $\left(Y_j^{F_i}\right)_{j \in \mathbb{N}}$ denote an i.i.d. sequence of random variables with distribution $F_i \in \mathcal{G}(\mu_i)$ satisfying Assumption 1. Let $\Theta \equiv \{\theta_m : m = 1, 2, ...\}$ be a deterministic non-negative real-valued sequence such that $\{(\theta_m/m) : m = 1, 2, ...\}$ is monotone decreasing in $m$ with $\theta_1 < 1$ and $\theta_m = o(m)$. Then,*

$$\beta_\Delta := \min_{F_1 \in \mathcal{G}(\mu_1), F_2 \in \mathcal{G}(\mu_2)} \mathbb{P}\left(\bigcap_{m=1}^{\infty} \left|\sum_{j=1}^{m}\left(Y_j^{F_1} - Y_j^{F_2}\right)\right| \geqslant \theta_m\right) > 0. \qquad (1)$$

**Proof of Proposition 1.** Refer to Appendix C (Note: Assumption 1 plays a key role here.). $\qquad \square$

**Remark.** $\beta_\Delta$ is a continuous function of $\Delta$ with $\lim_{\Delta \to 0} \beta_\Delta = 0$. In particular, $\beta_\Delta$ depends on $\Delta$ and the specific choice of $\Theta$. Proposition 1 implicitly assumes $\Delta > 0$.

**Theorem 3 (Upper bound on the expected regret of ALG(UCB1, $\Theta$, 2))** *Consider the input sequence $\Theta \equiv \{\theta_m : m = 1, 2, ...\}$ given by*

$$\theta_m := \sqrt{m^2(m + m_0)^{-1}\left(4\log(m + m_0) + \gamma \log\log(m + m_0)\right)}, \qquad (2)$$

*where $m_0 \geqslant 0$ and $\gamma > 2$ are user-defined parameters that ensure $\Theta$ satisfies the conditions of Proposition 1 (for example, $m_0 = 11$ and $\gamma = 2.1$ is an acceptable configuration). Suppose that Assumption 1 is satisfied. Then, the expected cumulative regret of $\pi = ALG(UCB1, \Theta, 2)$ after any number of plays $n$ is bounded as follows:*

$$\mathbb{E}R_n^\pi \leqslant \min\left(\Delta n, \, 8\left(\Delta\beta_\Delta\right)^{-1}\log n + \left(C_1 + \alpha^{-1}C_2\right)\beta_\Delta^{-1}\Delta\right), \qquad (3)$$

*where $\beta_\Delta$ is as defined in (1) with $\Theta$ specified via (2), $\Delta = \mu_1 - \mu_2 > 0$, $C_1$ is an absolute constant and $C_2$ is a constant that depends only on the free parameters of the algorithm, namely $(m_0, \gamma)$.*

**Comparison with the two-armed bandit problem.** The expected cumulative regret of $\pi = \text{UCB1}$ [5] after any number of plays $n$ in a two-armed bandit problem with gap $\Delta$ is bounded as follows:

$$\mathbb{E}R_n^\pi \leqslant \min\left(\Delta n,\ 8\Delta^{-1}\log n + C_1\Delta\right). \tag{4}$$

Observe that the upper bounds in (3) and (4) differ in $(\alpha, \beta_\Delta, C_2)$. The presence of the inflation factor $\beta_\Delta^{-1}$ in (3) is on account of the samples "wasted" due to false positives (rejecting the null, when it is in fact true) in the CAB problem. Specifically, $1 - \beta_\Delta$ is an upper bound on the false positive rate of $\text{ALG}(\text{UCB1}, \Theta, 2)$ (Proposition 1). Furthermore, $\beta_\Delta$ is invariant w.r.t. the playing rule (UCB1, in this case) as long as it is sufficiently exploratory (This statement is formalized in Lemma 1,2 stated in Appendix F.). In that sense, $\beta_\Delta$ captures the added layer of complexity due to the countable-armed extension of the finite-armed problem. We believe this is not merely an artifact of our proof but in fact, reflecting a fundamentally different scaling of the best achievable regret in the CAB problem vis-à-vis its finite-armed counterpart. It is also noteworthy that $\beta_\Delta$ is independent of $\alpha$; the implication is that (3) depends on the proportion of optimal arms only through the constant term, unlike Theorem 2.

**Dependence of $\beta_\Delta$ on $\Delta$.** Obtaining a closed-form expression for $\beta_\Delta$ as a function of $\Delta$ (cf. (1)) is not possible, we therefore resort to numerical evaluations using Monte-Carlo simulations.

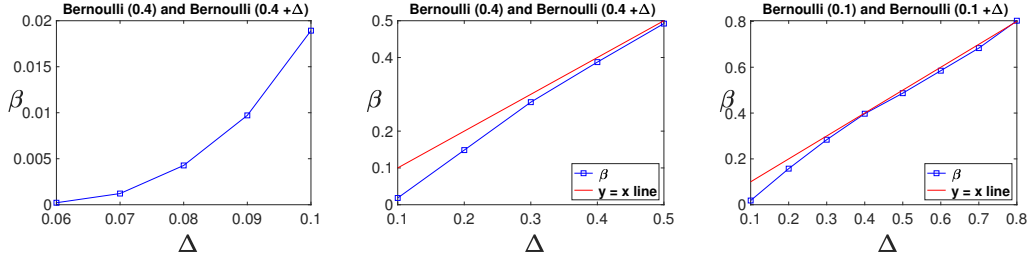

Figure 1: $\beta_\Delta$ vs. $\Delta$: Monte-Carlo estimates of $\beta_\Delta$ plotted against $\Delta$ using (2) with $m_0 = 4000$ and $\gamma = 2.1$. Rewards associated with each type $i \in \mathcal{T}$ are modeled as Bernoulli$(\mu_i)$.

An immediate observation from Figure 1 is that $\beta_\Delta \approx \Delta$ when $\Delta$ is sufficiently large (see center and rightmost plots). This has the implication that the upper bound of Theorem 3 scales approximately as $\mathcal{O}\left(\Delta^{-2}\log n\right)$ on well-separated instances, which can be contrasted with the classical $\mathcal{O}\left(\Delta^{-1}\log n\right)$ scaling achievable in finite-armed problems. The extra $\Delta^{-1}$ term is reflective of the additional complexity of the CAB problem vis-à-vis the finite-armed problem. In addition, for small $\Delta$ (see leftmost plot), $\beta_\Delta$ seems to vanish very fast as $\Delta \to 0$. This suggests that the minimax regret of $\text{ALG}(\text{UCB1}, \Theta, 2)$ is orders of magnitude larger (in $n$) than $\mathcal{O}\left(\sqrt{n\log n}\right)$, which is UCB1's minimax regret in finite-armed problems. Of course, characterizing the minimax statistical complexity of the CAB model and the design of algorithms that can achieve the best possible problem-independent rates, remain open problems at the moment.

**Significance of UCB1's concentration in zero gap.** That $C_2$ (appearing in (3)) is a constant is a highly non-trivial consequence of the concentration property of UCB1 à la part (i) of Theorem 4 stated in § 4. In the absence of this property, $C_2$ would scale with the horizon of play linearly and $\text{ALG}(\text{UCB1}, \Theta, 2)$ would effectively suffer linear regret. In what follows, we will demonstrate empirically that *Thompson Sampling most likely does not enjoy this concentration property*. To the best of our knowledge, this is the first example illustrating such a drastic performance disparity between algorithms based on UCB and Thompson Sampling in any stochastic bandit problem.

**Proof sketch of Theorem 3.** On homogeneous $\mathcal{A}$'s with arms of the optimal type (type 1), any playing rule incurs zero regret in expectation, whereas the expected regret is linear on homogeneous $\mathcal{A}$'s of type 2. On heterogeneous $\mathcal{A}$'s, the expected regret of UCB1 is logarithmic in the number of plays anytime. Since it is statistically impossible to distinguish between homogeneous $\mathcal{A}$'s of type 1 and type 2 in the absence of any distributional knowledge of the associated rewards, the decision maker must allocate all of her sampling effort (in expectation) to heterogeneous $\mathcal{A}$'s, to avoid high regret. This would ensure that UCB1 runs "uninterrupted" (in expectation) over the duration of play,

thereby guaranteeing logarithmic regret. This argument precisely forms the backbone of our proof. The number of re-initializations of the algorithm needed for a heterogeneous $\mathcal{A}$ to get selected is a geometric random variable and furthermore, every time a homogeneous $\mathcal{A}$ gets selected, the algorithm re-initializes within a finite number of plays in expectation. Therefore, only finitely many plays (in expectation) are spent on homogeneous $\mathcal{A}$'s until a heterogeneous $\mathcal{A}$ gets selected. Subsequently, the algorithm (in expectation) allocates the residual sampling effort to $\mathcal{A}$ which allows UCB1 to run uninterrupted, thereby guaranteeing logarithmic regret. Full proof is relegated to Appendix D. □

**Miscellaneous remarks. (i) Comparison with the state-of-the-art.** The regret incurred by suitable adaptations of known algorithms for infinite-armed bandits, e.g., [22], etc., is provably worse by at least poly-logarithmic factors compared to the optimal $\mathcal{O}(\log n)$ rate achievable in the CAB problem. **(ii) Alternatives to UCB1 in ALG(UCB1, $\Theta$, 2).** The choice of UCB1 is entirely a consequence of our desire to keep the analysis simple, and does not preclude use of suitable alternatives satisfying a concentration property akin to part (i) of Theorem 4. **(iii) Improving sample-efficiency.** ALG(UCB1, $\Theta$, 2) indulges in wasteful exploration since it selects an entirely new consideration set of arms at the beginning of every epoch. This is done for the simplicity of analysis. Sample-efficiency can be improved by discarding only one arm at the end of an epoch and selecting only one new arm at the beginning of the next. Furthermore, sample history of the arm retained from the previous epoch can also be used in subsequent hypothesis testing iterations for faster identification of homogeneous consideration sets without forcing unnecessary additional plays. **(iv) Limitations.** In this paper, we assume that $|\mathcal{T}|$ is perfectly known to the decision maker. However, it remains unclear if sublinear regret would still be information-theoretically achievable on "well-separated" instances if said assumption is violated, ceteris paribus.

## 4   UCB1 and the zero gap problem

UCB1 [5] is a celebrated optimism-based algorithm for finite-armed bandits that adapts to the sub-optimality gap (separation) between the top two arms, and guarantees a worst-case regret of $\mathcal{O}(\sqrt{n \log n})$ (ignoring dependence on the number of arms). This occurs when the separation scales with the horizon of play as $\mathcal{O}\left(\sqrt{n^{-1} \log n}\right)$. Our interest here, however, concerns the scenario where this separation is exactly *zero*, as opposed to simply being vanishingly small in the limit $n \to \infty$. Of immediate consequence to our CAB model, we restrict our focus to the special case of a stochastic two-armed bandit with *equal* mean rewards. Regret related questions are irrelevant in this setting since every policy incurs zero regret in expectation. However, asymptotics of UCB1 and the sampling balance (or imbalance) between the arms in *zero gap*, remain poorly understood in extant literature[6] to the best of our knowledge. In this paper, we provide the first analysis in this direction.

**Theorem 4 (Concentration of UCB1 in zero gap)** *Consider a stochastic two-armed bandit with rewards bounded in $[0, 1]$ and arms having equal means. Let $N_i(n)$ denote the number of plays of arm $i$ under UCB1 [5] up to and including time $n$. Then, the following results hold for any $i \in \{1, 2\}$:*

*(i)* **Concentration.** *For any $n \in \mathbb{N}$ and $\epsilon \in (0, 1/2)$,*

$$\mathbb{P}\left(\left|\frac{N_i(n)}{n} - \frac{1}{2}\right| > \epsilon\right) < 8n^{-\left(3 - 4\sqrt{1 - 4\epsilon^2}\right)}.$$

*(ii)* **Convergence.** *$N_i(n)/n \to 1/2$ in probability as $n \to \infty$ (Convergence does not follow from concentration alone since the bound in (i) is vacuous for $\epsilon \leqslant \sqrt{7}/8$.)*

**Result for generic UCB.** Theorem 4 also extends to the generic UCB policy that uses $\sqrt{\rho n^{-1} \log n}$ as the optimistic bias, where $\rho > 1/2$ is called the exploration coefficient ($\rho = 2$ corresponds to UCB1). The concentration bound for said policy (informally called UCB($\rho$)) is given by

$$\mathbb{P}\left(\left|\frac{N_i(n)}{n} - \frac{1}{2}\right| > \epsilon\right) < 2^{2\rho - 1} n^{-\left(2\rho - 1 - 2\rho\sqrt{1 - 4\epsilon^2}\right)}. \tag{5}$$

While the tail progressively gets lighter as $\rho$ increases, it is achieved at the expense of an inflated regret on instances with non-zero gap. Specifically, the authors in [4] showed that the expected

regret of UCB($\rho$) on well-separated instances scales as $\mathcal{O}(\rho \log n)$. They also showed that the tail of UCB($\rho$)'s pseudo-regret on well-separated instances is bounded as $\mathbb{P}(R_n > z) = \mathcal{O}\left(z^{-(2\rho-1)}\right)$ for large enough $z$, implying a tail decay of $\mathcal{O}\left(z^{-(2\rho-1)}\right)$ for the fraction of *inferior* plays. On the other hand, (5) suggests for the fractional plays of *any* arm, a heavier tail decay of $\mathcal{O}\left(z^{-\left(2\rho-1-2\rho\sqrt{1-4\epsilon^2}\right)}\right)$ in zero gap settings, which accounts for the slow convergence evident in Figure 2 (leftmost plot).

**Miscellaneous remark.** Theorem 4 (the convergence result in part (ii), in particular) is likely to have implications for inference problems involving adaptive data collected by UCB-inspired algorithms.

**Parsing Theorem 4.** To build some intuition, we pivot to the case of statistically identical arms. In this case, labels are exchangeable and therefore $\mathbb{E}(N_i(n)/n) = 1/2$ for $i \in \{1,2\}, n \in \mathbb{N}$. While symmetry between the arms is enough to guarantee convergence in expectation, it does not shed light on the pathwise behavior of UCB1. An immediate corollary of part (i) of Theorem 4 is that for any $\epsilon \in \left(\sqrt{3}/4, 1/2\right)$ and $i \in \{1,2\}$, it so happens that $\sum_{n \in \mathbb{N}} \mathbb{P}\left(|N_i(n)/n - 1/2| > \epsilon\right) < \infty$. The Borel-Cantelli lemma then implies that the arms are eventually sampled linearly in time, almost surely, at a rate that is at least $\left(1/2 - \sqrt{3}/4\right)$. That this rate cannot be pushed arbitrarily close to $1/2$ is not merely an artifact of our proof but also suggested by the extremely slow convergence of the empirical probability density of $N_1(n)/n$ to the Dirac delta at $1/2$ in Figure 2 (leftmost plot). This slow convergence likely led to the incorrect folk conjecture that optimism-based algorithms such as UCB1 and variants thereof do not converge à la part (ii) of Theorem 4 (e.g., see [14] and references therein). Instead, we believe the weaker conjecture that the convergence is not w.p. 1, is likely true. Full proof is given in Appendix E.

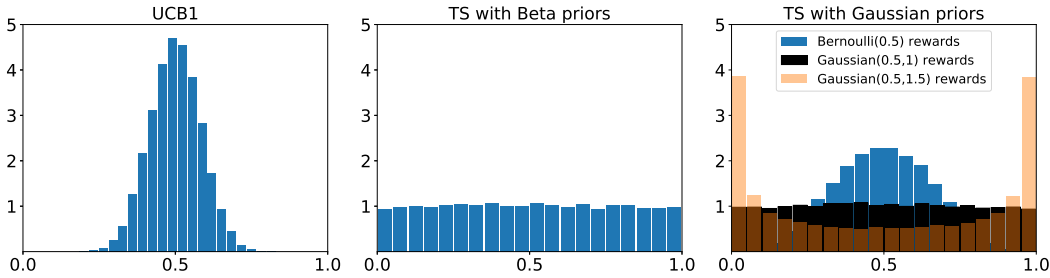

Figure 2: Two-armed bandit with Bernoulli$(0.5)$ rewards: Histogram of the fraction of plays of arm 1 until time $n = 10{,}000$, i.e., $N_1\left(10^4\right)/10^4$, under three different algorithms. Number of replications under each algorithm $\aleph = 20{,}000$. The algorithms are: UCB1 (leftmost), Thompson Sampling (TS) with Beta priors (center) and TS with Gaussian priors (rightmost) [3]. The last plot shows histograms for 3 reward configurations: Bernoulli$(0.5)$ (blue), $\mathcal{N}(0.5, 1)$ (dashed), and $\mathcal{N}(0.5, 1.5)$ (orange).

**Empirical illustration.** Figure 2 shows the histogram of the fraction of time a particular arm of a two-armed bandit having statistically identical arms with Bernoulli$(0.5)$ rewards each was played under different algorithms. The leftmost plot corresponds to UCB1 and is evidently in consonance with the concentration property stated in part (i) of Theorem 4. The concentration phenomenon under UCB1 can be understood through the lens of reward stochasticity. Consider the simplest case where the rewards are deterministic. Then, we know from the structure of UCB1 that any arm is played at most twice before the algorithm switches over to the other arm. This results in $N_1(n)/n$ converging to $1/2$ pathwise, with an arm switch-over time that is at most 2. As the reward stochasticity increases, so does the arm switch-over time, which adversely affects this convergence. While it is a priori unclear whether $N_1(n)/n$ would still converge to $1/2$ in some mode if the rewards are stochastic, part (ii) of Theorem 4 states that the convergence indeed holds, albeit only in probability. A significant spread around $1/2$ in the leftmost plot despite $n = 10^4$ plays indicates a rather slow convergence.

**A remark on Thompson Sampling.** Concentration and convergence à la Theorem 4 should be contrasted with other popular gap-agnostic algorithms such as Thompson Sampling (TS). Empirical evidence suggests that the behavior of TS is drastically different from UCB1's in zero gap problems (see Figure 2). Furthermore, there seems to be a fundamental difference even between different TS instantiations. While a conjectural Uniform$(0, 1)$ limit may be rationalized by Proposition 1 in [23], understanding the trichotomy in the rightmost plot and its implications remains an open problem.

## Broader Impact

The authors do not claim any immediate broader impact of this work as such.

## Acknowledgments and Disclosure of Funding

The authors thank the anonymous referees for their constructive feedback on the initial version of this paper. The authors also declare an absence of any competing interests.

## Footnotes

[1]This is simply to keep the analysis simple and has no bearing on the regret guarantees of our algorithms.

[2]Define $\lambda(F_i, F_j) := \max_{(k,l) \in \{(i,j),(j,i)\}} (\inf\{x \in \mathbb{R} : F_k(x) = 1\} - \sup\{x \in \mathbb{R} : F_l(x) = 0\})$ for arbitrary CDFs $F_i, F_j$. We require prior knowledge of $\lambda_0 := \min_{i,j \in \mathcal{T}, i \neq j} \min_{F_i \in \mathcal{G}(\mu_i), F_j \in \mathcal{G}(\mu_j)} \lambda(F_i, F_j)$. Assumption 1 fixes $\lambda_0 = 1$.

[3]Expected cumulative regret equals the expected cumulative pseudo-regret in the stochastic bandits setting.

[4]This is a rich policy class that includes all algorithms achieving sublinear regret (defined in Appendix A).

[5]The standard exponential doubling trick can be employed to make the algorithm horizon-free, cf. [8].

[6]Extant work assumes a positive gap (cf. [4]); the resulting bounds are vacuous in the zero gap regime.

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
