[Supplementary Material]

# From Finite to Countable-Armed Bandits: Appendix

**Anand Kalvit[1] and Assaf Zeevi[2]**
Graduate School of Business
Columbia University
New York, USA
{[1]akalvit22,[2]assaf}@gsb.columbia.edu

## A  Proof of Theorem 1

Since the horizon of play is fixed at $n$, the decision maker may play at most $n$ distinct arms. Therefore, it suffices to focus only on the sequence of the first $n$ arms that may be played. A *realization* of an instance $\nu = (\mathcal{G}(\mu_1), \mathcal{G}(\mu_2))$ is defined as the $n$-tuple $r \equiv (r_i)_{1 \leqslant i \leqslant n}$, where $r_i \in \mathcal{G}(\mu_1) \cup \mathcal{G}(\mu_2)$ indicates the reward distribution of arm $i \in \{1, 2, ..., n\}$. It must be noted that the decision maker need not play every arm in $r$. The distribution over the possible realizations of $\nu = (\mathcal{G}(\mu_1), \mathcal{G}(\mu_2))$ in $\{r : r_i \in \mathcal{G}(\mu_1) \cup \mathcal{G}(\mu_2),\ 1 \leqslant i \leqslant n\}$ satisfies $\mathbb{P}(r_i \in \mathcal{G}(\max(\mu_1, \mu_2)) = \alpha$ for all $i \in \{1, 2, ..., n\}$.

Recall that the cumulative pseudo-regret after $n$ plays of a policy $\pi$ on $\nu = (\mathcal{G}(\mu_1), \mathcal{G}(\mu_2))$ is given by $R_n^\pi(\nu) = \sum_{m=1}^n \left(\max(\mu_1, \mu_2) - \mu_{t(\pi_m)}\right)$, where $t(\pi_m) \in \{1, 2\}$ indicates the type of the arm played by $\pi$ at time $m$. Our goal is to lower bound $\mathbb{E}R_n^\pi(\nu)$, where the expectation is w.r.t. the randomness in $\pi$ as well as the distribution over the possible realizations of $\nu$. To this end, we define the notion of expected cumulative regret of $\pi$ on a realization $r$ of $\nu = (\mathcal{G}(\mu_1), \mathcal{G}(\mu_2))$ by

$$S_n^\pi(\nu, r) := \mathbb{E}^\pi \left[ \sum_{m=1}^n \left(\max(\mu_1, \mu_2) - \mu_{t(\pi_m)}\right) \right],$$

where the expectation $\mathbb{E}^\pi$ is w.r.t. the randomness in $\pi$. Note that $\mathbb{E}R_n^\pi(\nu) = \mathbb{E}^\nu S_n^\pi(\nu, r)$, where the expectation $\mathbb{E}^\nu$ is w.r.t. the distribution over the possible realizations of $\nu$. We define our problem class $\mathcal{N}_\Delta$ as the collection of $\Delta$-separated instances given by

$$\mathcal{N}_\Delta := \left\{ (\mathcal{G}(\mu_1), \mathcal{G}(\mu_2)) : \mu_1 - \mu_2 = \Delta,\ (\mu_1, \mu_2) \in \mathbb{R}^2 \right\}.$$

**Definition 1 (Consistent policy)** *Let $\Lambda(r)$ denote the number of "optimal" arms in realization $r$. We call $\pi$, an asymptotically consistent policy for the problem class $\mathcal{N}_\Delta$ if for any instance $\nu \in \mathcal{N}_\Delta$ and any realization $r$ thereof, it satisfies the following two conditions:*

$$\mathbb{E}R_n^\pi(\nu) = o(n^p) \qquad\qquad \text{for every } p \in (0, 1),\ \alpha \in (0, 1]. \quad (1)$$
$$\mathbb{E}^\nu \left[ S_n^\pi(\nu, r) \big| \Lambda(r) = m \right] \geqslant \mathbb{E}^\nu \left[ S_n^\pi(\nu, r) \big| \Lambda(r) = k \right] \qquad \forall\, (m, n, k) : 0 \leqslant m \leqslant k \leqslant n. \quad (2)$$

The set of such policies is denoted by $\Pi_{\text{cons}}(\mathcal{N}_\Delta)$. Notice that (1), barring the condition on $\alpha$, is the standard definition of asymptotic consistency first introduced in [6] and subsequently adopted by many other papers. The exclusion of $\alpha = 0$ is necessary since no policy can achieve sublinear regret in said case. We also remark that the additional condition in (2) is not restrictive since any reasonable policy is expected to incur a larger cumulative regret (in expectation) on realizations with fewer optimal arms.

Fix an arbitrary $\Delta > 0$ and consider an instance $\nu = (\{Q_1\}, \{Q_2\}) \in \mathcal{N}_\Delta$, where $(Q_1, Q_2)$ are unit-variance Gaussian distributions with means $(\mu_1, \mu_2)$ respectively. Consider an arbitrary realization $r \in \{Q_1, Q_2\}^n$ of $\nu$ and let $\mathcal{I} \subseteq \{1, 2, ..., n\}$ denote the set of inferior arms in $r$ (arms with reward distribution $Q_2$). Consider another instance $\nu' \in \mathcal{N}_\Delta$ given by $\nu' = \left( \{\widetilde{Q}_1\}, \{Q_1\} \right)$, where $\widetilde{Q}_1$ is

another unit variance Gaussian with mean $\mu_1 + \Delta$. Now consider a realization $r' \in \{\widetilde{Q}_1, Q_1\}^n$ of $\nu'$ that is such that the arms at positions in $\mathcal{I}$ have distribution $\widetilde{Q}_1$ while those at positions in $\{1, 2, ..., n\} \backslash \mathcal{I}$ have distribution $Q_1$. Notice that $\mathcal{I}$ is the set of optimal arms in $r'$ (arms with reward distribution $\widetilde{Q}_1$), implying $\Lambda(r') = |\mathcal{I}|$. Then, the following always holds:

$$S_n^\pi(\nu, r) + S_n^\pi(\nu', r') \geqslant \left(\frac{\Delta n}{2}\right) \left(\mathbb{P}_{\nu, r}^\pi \left(\sum_{i \in \mathcal{I}} N_i(n) > \frac{n}{2}\right) + \mathbb{P}_{\nu', r'}^\pi \left(\sum_{i \in \mathcal{I}} N_i(n) \leqslant \frac{n}{2}\right)\right),$$

where $\mathbb{P}_{\nu, r}^\pi(\cdot)$ and $\mathbb{P}_{\nu', r'}^\pi(\cdot)$ denote the probability measures w.r.t. the instance-realization pairs $(\nu, r)$ and $(\nu', r')$ respectively, and $N_i(n)$ denotes the number of plays up to and including time $n$ of arm $i \in \{1, 2, ..., n\}$. Using the Bretagnolle-Huber inequality (Theorem 14.2 of [7]), we obtain

$$S_n^\pi(\nu, r) + S_n^\pi(\nu', r') \geqslant \left(\frac{\Delta n}{4}\right) \exp\left(-\mathrm{D}\left(\mathbb{P}_{\nu, r}^\pi, \mathbb{P}_{\nu', r'}^\pi\right)\right),$$

where $\mathrm{D}\left(\mathbb{P}_{\nu, r}^\pi, \mathbb{P}_{\nu', r'}^\pi\right)$ denotes the KL-Divergence between $\mathbb{P}_{\nu, r}^\pi$ and $\mathbb{P}_{\nu', r'}^\pi$. Using Divergence decomposition (Lemma 15.1 of [7]), we further obtain

$$S_n^\pi(\nu, r) + S_n^\pi(\nu', r') \geqslant \left(\frac{\Delta n}{4}\right) \exp\left(-\left(\frac{\mathrm{D}\left(Q_2, \widetilde{Q}_1\right)}{\Delta}\right) S_n^\pi(\nu, r)\right) = \left(\frac{\Delta n}{4}\right) \exp\left(-2\Delta S_n^\pi(\nu, r)\right),$$

where the equality follows since $\widetilde{Q}_1$ and $Q_2$ are unit variance Gaussian distributions with means separated by $2\Delta$. Next, taking the expectation $\mathbb{E}^\nu$ on both the sides above and a direct application of Jensen's inequality thereafter yields

$$\mathbb{E}R_n^\pi(\nu) + \mathbb{E}^\nu S_n^\pi(\nu', r') \geqslant \left(\frac{\Delta n}{4}\right) \exp\left(-2\Delta\mathbb{E}R_n^\pi(\nu)\right). \tag{3}$$

Consider the $\mathbb{E}^\nu S_n^\pi(\nu', r')$ term in (3) and an arbitrary $\alpha \in (0, 1/2]$. Using a simple change-of-measure argument, we obtain

$$\mathbb{E}^\nu S_n^\pi(\nu', r') = \mathbb{E}^{\nu'} \left[S_n^\pi(\nu', r') \left(\frac{1-\alpha}{\alpha}\right)^{2\left(\Lambda(r') - n/2\right)}\right]$$

$$\leqslant \mathbb{E}R_n^\pi(\nu') + \mathbb{E}^{\nu'} \left[S_n^\pi(\nu', r') \left(\frac{1-\alpha}{\alpha}\right)^{2\left(\Lambda(r') - n/2\right)} \mathbb{1}\left\{\Lambda(r') > n/2\right\}\right], \tag{4}$$

where the inequality follows since $\alpha \leqslant 1/2$. Now consider the second term on the RHS in (4). It follows that

$$\mathbb{E}^{\nu'} \left[S_n^\pi(\nu', r') \left(\frac{1-\alpha}{\alpha}\right)^{2\left(\Lambda(r') - n/2\right)} \mathbb{1}\left\{\Lambda(r') > n/2\right\}\right]$$

$$= \sum_{k > n/2} \mathbb{E}^{\nu'} \left[S_n^\pi(\nu', r') \left(\frac{1-\alpha}{\alpha}\right)^{2\left(\Lambda(r') - n/2\right)} \mathbb{1}\left\{\Lambda(r') = k\right\}\right]$$

$$= \sum_{k > n/2} \left(\frac{1-\alpha}{\alpha}\right)^{(2k-n)} \mathbb{E}^{\nu'} \left[S_n^\pi(\nu', r') \mathbb{1}\left\{\Lambda(r') = k\right\}\right]$$

$$= \sum_{k > n/2} \left(\frac{1-\alpha}{\alpha}\right)^{(2k-n)} \mathbb{E}^{\nu'} \left[S_n^\pi(\nu', r') | \Lambda(r') = k\right] \mathbb{P}_{\nu'}\left(\Lambda(r') = k\right)$$

$$= \sum_{k > n/2} \left(\frac{1-\alpha}{\alpha}\right)^{(2k-n)} \mathbb{E}^{\nu'} \left[S_n^\pi(\nu', r') | \Lambda(r') = k\right] \binom{n}{k} \alpha^k (1-\alpha)^{(n-k)}$$

$$= \alpha^n \sum_{k > n/2} \binom{n}{k} \left(\frac{1-\alpha}{\alpha}\right)^k \mathbb{E}^{\nu'} \left[S_n^\pi(\nu', r') | \Lambda(r') = k\right]. \tag{5}$$

Recall that $\nu' \in \mathcal{N}_\Delta$ and $\pi \in \Pi_{\text{cons}}(\mathcal{N}_\Delta)$. We have

$$\mathbb{E}R_n^\pi(\nu') = \mathbb{E}^{\nu'} S_n^\pi(\nu', r')$$

$$\geqslant \sum_{m=1}^k \mathbb{E}^{\nu'} \left[ S_n^\pi(\nu', r') \middle| \Lambda(r') = m \right] \mathbb{P}_{\nu'}(\Lambda(r') = m) \qquad \text{(for any } k \leqslant n\text{)}$$

$$\geqslant \mathbb{E}^{\nu'} \left[ S_n^\pi(\nu', r') \middle| \Lambda(r') = k \right] \mathbb{P}_{\nu'}(\Lambda(r') \leqslant k). \qquad \text{(using (2))} \qquad (6)$$

Since $\alpha \leqslant 1/2$, it follows that for any $k > n/2$, $\mathbb{P}_{\nu'}(\Lambda(r') \leqslant k) = \mathcal{O}_n(1)$ (the subscript $n$ indicates that the asymptotic scaling is w.r.t. $n$). Using this observation together with (1) and (6), we conclude that

$$\forall\, k > n/2,\ \alpha \in (0, 1/2] \text{ and every } p \in (0,1),\ \mathbb{E}^{\nu'}\left[ S_n^\pi(\nu', r') \middle| \Lambda(r') = k \right] = o\left(n^p\right). \qquad (7)$$

Combining (4), (5), (7) and using the fact that $\nu' \in \mathcal{N}_\Delta$ with $\pi \in \Pi_{\text{cons}}(\mathcal{N}_\Delta)$, we conclude

$$\forall\, k > n/2,\ \alpha \in (0, 1/2] \text{ and every } p \in (0,1),\ \mathbb{E}^\nu S_n^\pi(\nu', r') = o\left(n^p\right). \qquad (8)$$

Now consider (3). Taking the natural logarithm of both sides and rearranging, we obtain

$$\frac{\mathbb{E}R_n^\pi(\nu)}{\log n} \geqslant \left(\frac{1}{2\Delta}\right)\left(1 + \frac{\log\left(\frac{\Delta}{4}\right)}{\log n} - \frac{\log(\mathbb{E}R_n^\pi(\nu) + \mathbb{E}^\nu S_n^\pi(\nu', r'))}{\log n}\right).$$

Since $\nu, \nu' \in \mathcal{N}_\Delta$ and $\pi \in \Pi_{\text{cons}}(\mathcal{N}_\Delta)$, the assertion follows using (8) that for any $\alpha \in (0, 1/2]$,

$$\liminf_{n \to \infty} \frac{\mathbb{E}R_n^\pi(\nu)}{\log n} \geqslant \frac{1}{2\Delta}.$$

Therefore, for any $\Delta > 0$, $\exists\, \nu \in \mathcal{N}_\Delta$ and an absolute constant $C$ s.t. the expected cumulative regret of any consistent policy $\pi$ on $\nu$ satisfies $\forall\, \alpha \leqslant 1/2$ and $n$ large enough, $\mathbb{E}R_n^\pi(\nu) \geqslant C\Delta^{-1} \log n$. $\square$

## B  Proof of Theorem 2

We divide the horizon of play into epochs of length $m$ each. For each $k \geqslant 0$, let $S_k$ denote the cumulative pseudo-regret incurred by the algorithm when it is initialized at the beginning of epoch $(2k+1)$ and continued until the end of the horizon of play, i.e., the algorithm starts at time $2km + 1$ and runs until time $n$. We are interested in an upper bound on $\mathbb{E}R_n^\pi = \mathbb{E}S_0$. To this end, suppose that the algorithm is initialized at time $2km + 1$. Label the arms played in epochs $(2k+1)$ and $(2k+2)$ as '1' and '2' respectively. Let $\overline{X}_i$ denote the empirical mean reward from $m$ plays of arm $i \in \{1, 2\}$. Recall that $t(i) \in \mathcal{T} = \{1, 2\}$ denotes the type of arm $i$, that type 1 is assumed optimal and lastly, that the probability of a new arm being of the optimal type is $\alpha$. Suppose that $\mathbb{1}\{E\}$ denotes the indicator random variable associated with event $E$. Then, we have that $S_k$ evolves according to the following stochastic recursive relation:

$$S_k = \mathbb{1}\{t(1) = 1, t(2) = 2\} \left[ \Delta m + \mathbb{1}\{\overline{X}_2 - \overline{X}_1 > \delta\} \left[n - (2k+2)m\right] + \mathbb{1}\{|\overline{X}_1 - \overline{X}_2| < \delta\} S_{k+1} \right] +$$

$$\mathbb{1}\{t(1) = 2, t(2) = 1\} \left[ \Delta m + \mathbb{1}\{\overline{X}_1 - \overline{X}_2 > \delta\} \left[n - (2k+2)m\right] + \mathbb{1}\{|\overline{X}_1 - \overline{X}_2| < \delta\} S_{k+1} \right] +$$

$$\mathbb{1}\{t(1) = 2, t(2) = 2\} \left[ 2\Delta m + \mathbb{1}\{|\overline{X}_1 - \overline{X}_2| > \delta\}\Delta \left[n - (2k+2)m\right] + \mathbb{1}\{|\overline{X}_1 - \overline{X}_2| < \delta\} S_{k+1} \right] +$$

$$\mathbb{1}\{t(1) = 1, t(2) = 1\}\mathbb{1}\{|\overline{X}_1 - \overline{X}_2| < \delta\} S_{k+1}.$$

Collecting like terms together,

$$S_k = \mathbb{1}\{t(1) = 1, t(2) = 2\}\mathbb{1}\{\overline{X}_2 - \overline{X}_1 > \delta\}\Delta \left[n - (2k+2)m\right] +$$

$$\mathbb{1}\{t(1) = 2, t(2) = 1\}\mathbb{1}\{\overline{X}_1 - \overline{X}_2 > \delta\}\Delta \left[n - (2k+2)m\right] +$$

$$\mathbb{1}\{t(1) = 2, t(2) = 2\}\mathbb{1}\{|\overline{X}_1 - \overline{X}_2| > \delta\}\Delta \left[n - (2k+2)m\right] +$$

$$\left[\mathbb{1}\{t(1) \neq t(2)\} + 2\mathbb{1}\{t(1) = 2, t(2) = 2\}\right]\Delta m + \mathbb{1}\{|\overline{X}_1 - \overline{X}_2| < \delta\} S_{k+1}. \qquad (9)$$

Define the following conditional events:

$$E_1 := \left\{ \overline{X}_2 - \overline{X}_1 > \delta \;\middle|\; t(1) = 1,\ t(2) = 2 \right\}, \qquad (10)$$

$$E_2 := \left\{ \overline{X}_1 - \overline{X}_2 > \delta \;\middle|\; t(1) = 2,\ t(2) = 1 \right\}, \qquad (11)$$

$$E_3 := \left\{ \left|\overline{X}_1 - \overline{X}_2\right| > \delta \;\middle|\; t(1) = 2,\ t(2) = 2 \right\}, \qquad (12)$$

$$E_4 := \left\{ \left|\overline{X}_1 - \overline{X}_2\right| < \delta \;\middle|\; t(1) = t(2) \right\}, \qquad (13)$$

$$E_5 := \left\{ \left|\overline{X}_1 - \overline{X}_2\right| < \delta \;\middle|\; t(1) \neq t(2) \right\}. \qquad (14)$$

Taking expectations on both sides in (9) and rearranging, one obtains the following using (10),(11),(12),(13),(14):

$$
\begin{aligned}
\mathbb{E}S_k = {} & \left[\alpha(1-\alpha)\left\{\mathbb{P}(E_1)+\mathbb{P}(E_2)\right\}+(1-\alpha)^2\mathbb{P}(E_3)\right]\Delta\left[n-(2k+2)m\right] \\
& + \left[2\alpha(1-\alpha)+2(1-\alpha)^2\right]\Delta m + \mathbb{P}\left(\left|\overline{X}_1-\overline{X}_2\right|<\delta\right)\mathbb{E}S_{k+1}.
\end{aligned}
\tag{15}
$$

Notice that $S_{k+1}$, by definition, is independent of $(X_{i,j})_{i\in\{1,2\},1\leqslant j\leqslant m}$, and hence $\mathbb{E}\left[\mathbb{1}\{\left|\overline{X}_1-\overline{X}_2\right|<\delta\}S_{k+1}\right] = \mathbb{P}\left(\left|\overline{X}_1-\overline{X}_2\right|<\delta\right)\mathbb{E}S_{k+1}$ in (15). Further note that

$$
\mathbb{P}\left(\left|\overline{X}_1-\overline{X}_2\right|<\delta\right) = \left[\alpha^2+(1-\alpha)^2\right]\mathbb{P}(E_4)+2\alpha(1-\alpha)\mathbb{P}(E_5).
\tag{16}
$$

From (15) and (16), we conclude after a little rearrangement the following:

$$
\mathbb{E}S_k = \xi_1 - \xi_2 k + \xi_3\mathbb{E}S_{k+1},
\tag{17}
$$

where the $\xi_i$'s do not depend on $k$ and are given by

$$
\xi_1 := \Delta\left[\alpha(1-\alpha)\left\{\mathbb{P}(E_1)+\mathbb{P}(E_2)\right\}+(1-\alpha)^2\mathbb{P}(E_3)\right](n-2m)+2\Delta(1-\alpha)m,
\tag{18}
$$

$$
\xi_2 := 2\Delta\left[\alpha(1-\alpha)\left\{\mathbb{P}(E_1)+\mathbb{P}(E_2)\right\}+(1-\alpha)^2\mathbb{P}(E_3)\right]m,
\tag{19}
$$

$$
\xi_3 := \left[\alpha^2+(1-\alpha)^2\right]\mathbb{P}(E_4)+2\alpha(1-\alpha)\mathbb{P}(E_5).
\tag{20}
$$

Observe that the recursion in (17) is solvable in closed-form and admits the following solution:

$$
\mathbb{E}S_0 = \xi_1\sum_{k=0}^{l-1}\xi_3^k - \xi_2\sum_{k=0}^{l-1}k\xi_3^k + \xi_3^l\mathbb{E}S_l,
\tag{21}
$$

where $l := \lfloor n/(2m)\rfloor$. Since the $\xi_i$'s are all non-negative for $n\geqslant 2m$ and $\mathbb{E}S_l\leqslant 2\Delta m$, we have for $n\geqslant 2m$,

$$
\mathbb{E}R_n^\pi = \mathbb{E}S_0 \leqslant \frac{\xi_1}{1-\xi_3}+2\Delta m.
\tag{22}
$$

Now using (10),(11),(12),(13),(14) and Hoeffding's inequality [4] along with the fact that the $X_i$'s are bounded in $[0,1]$, we conclude

$$
\left\{\mathbb{P}(E_1),\mathbb{P}(E_2)\right\} \leqslant \exp\left(-(\Delta+\delta)^2 m/2\right),
\tag{23}
$$

$$
\left\{\mathbb{P}(E_3),\mathbb{P}(E_4^c)\right\} \leqslant 2\exp\left(-\delta^2 m/2\right),
\tag{24}
$$

$$
\mathbb{P}(E_5) \leqslant \exp\left(-(\Delta-\delta)^2 m/2\right).
\tag{25}
$$

From (18),(19),(20),(22),(23),(24) and (24), we conclude

$$
\mathbb{E}R_n^\pi \leqslant \frac{2\Delta n\exp\left(-\delta^2 m/2\right)+\Delta m}{\alpha\left(1-\exp\left(-(\Delta-\delta)^2 m/2\right)\right)}+2\Delta m.
$$

Finally since $m = \left\lceil(2/\delta^2)\log n\right\rceil$, the stated assertion follows, i.e., for all $n\geq 2m$,

$$
\mathbb{E}R_n^\pi \leqslant 2\Delta\left(1+\frac{1}{2\alpha}\right)\left[\left(\frac{2}{\delta^2}\right)\log n+1\right]+\left(\frac{\Delta}{\alpha}\right)\left[2+f(n,\delta,\Delta)\right],
\tag{26}
$$

where $f(n,\delta,\Delta)=o(1)$ in $n$ given by

$$
f(n,\delta,\Delta) := \left(\frac{n^{-\left(\frac{\Delta-\delta}{\delta}\right)^2}}{1-n^{-\left(\frac{\Delta-\delta}{\delta}\right)^2}}\right)\left[\left(\frac{2}{\delta^2}\right)\log n+3\right].
\tag{27}
$$

For $n<2m$, $\mathbb{E}R_n^\pi\leqslant 2\Delta m$ follows trivially. Therefore, the bound in (26) is valid for all $n\geqslant 1$. Of course, $\mathbb{E}R_n^\pi\leqslant\Delta n$ offers a sharper bound whenever $\Delta$ is very small, similar to finite-armed settings. Thus in conclusion, $\mathbb{E}R_n^\pi$ is bounded as follows for any $n$:

$$
\mathbb{E}R_n^\pi \leqslant \min\left[\Delta n,\ 2\Delta\left(1+\frac{1}{2\alpha}\right)\left\{\left(\frac{2}{\delta^2}\right)\log n+1\right\}+\left(\frac{\Delta}{\alpha}\right)\left\{2+f(n,\delta,\Delta)\right\}\right].
$$

$\square$

## C Proof of Proposition 1

The statement of the proposition assumes $|\mu_1 - \mu_2| = \Delta > 0$. However, we will only prove it for the case where $\mu_1 - \mu_2 = \Delta > 0$. The proof for the other case is symmetric and an identical bound will follow. Fix an arbitrary $(F_1, F_2) \in \mathcal{G}(\mu_1) \times \mathcal{G}(\mu_2)$ and consider the following stopping time:

$$\tau := \inf \left\{ n \geqslant 1 : \sum_{k=1}^{n} (\Psi_k - \bar{\theta}_n) < 0 \right\}, \tag{28}$$

where $\Psi_k := Y_{1,j}^{F_1} - Y_{2,j}^{F_2}$ and $\bar{\theta}_n := \theta_n/n$. Note that $\mathbb{E}\Psi_k = \Delta > 0$ (by assumption). Then, it follows that $\mathbb{P}\left( \bigcap_{m=1}^{\infty} \left| \sum_{j=1}^{m} \left( Y_{1,j}^{F_1} - Y_{2,j}^{F_2} \right) \right| \geqslant \theta_m \right) \geqslant \mathbb{P}(\tau = \infty)$. Therefore, it suffices to show that $\mathbb{P}(\tau = \infty)$ is bounded away from 0. To this end, fix an arbitrary $\lambda \in (0,1)$ and let $n_0 := \min\{k \in \mathbb{N} : \bar{\theta}_n \leqslant \lambda\Delta\}$. Since $\bar{\theta}_n \to 0$ as $n \to \infty$ and $\Delta > 0$, it follows that $n_0 < \infty$. Suppose that $\omega$ denotes an arbitrary sample-path and consider the following set:

$$E := \left\{ \omega : \Psi_k(\omega) > \bar{\theta}_k; \ 1 \leqslant k \leqslant n_0 \right\}. \tag{29}$$

Since Assumption 1 (main text) is satisfied, $n_0 < \infty$ and $\bar{\theta}_n$ is monotone decreasing in $n$ with $\bar{\theta}_1 < 1$, it follows that $\mathbb{P}(E)$, as given below, is strictly positive.

$$\mathbb{P}(E) = \prod_{k=1}^{n_0} \mathbb{P}\left( \Psi_k > \bar{\theta}_k \right) > 0, \ \text{where } n_0 = \min\{k \in \mathbb{N} : \bar{\theta}_n \leqslant \lambda\Delta\}. \tag{30}$$

Notice that $\tau > n_0$ on the event indicated by $E$. In particular,

$$\tau | E = \inf \left\{ n \geqslant n_0 + 1 : \sum_{k=n_0+1}^{n} (\Psi_k - \bar{\theta}_n) < -\sum_{k=1}^{n_0} (\Psi_k - \bar{\theta}_n) \ \middle| \ E \right\}$$

$$\underset{(\dagger)}{\geqslant} \inf \left\{ n \geqslant n_0 + 1 : \sum_{k=n_0+1}^{n} (\Psi_k - \bar{\theta}_n) < -\sum_{k=1}^{n_0} (\bar{\theta}_k - \bar{\theta}_n) \ \middle| \ E \right\}$$

$$\underset{(\ddagger)}{\geqslant} \inf \left\{ n \geqslant n_0 + 1 : \sum_{k=n_0+1}^{n} (\Psi_k - \bar{\theta}_n) < -\sum_{k=1}^{n_0} (\bar{\theta}_k - \bar{\theta}_{n_0}) \ \middle| \ E \right\}$$

$$\underset{(\bullet)}{\geqslant} \inf \left\{ n \geqslant n_0 + 1 : \sum_{k=n_0+1}^{n} (\Psi_k - \lambda\Delta) < -\sum_{k=1}^{n_0} (\bar{\theta}_k - \bar{\theta}_{n_0}) \ \middle| \ E \right\}$$

$$\underset{(\star)}{=} n_0 + \inf \left\{ n \geqslant 1 : \sum_{k=1}^{n} (\Psi_k' - \lambda\Delta) < -\eta \right\}, \tag{31}$$

where ($\dagger$) follows from (29), ($\ddagger$) follows since $\bar{\theta}_n \leqslant \bar{\theta}_{n_0}$ for $n \geqslant n_0$, ($\bullet$) since $\bar{\theta}_n \leqslant \lambda\Delta$ for $n \geqslant n_0$, and ($\star$) holds with $\eta := \sum_{k=1}^{n_0} (\bar{\theta}_k - \bar{\theta}_{n_0})$ and $\Psi_k' := \Psi_{n_0+k}$ since $(\Psi_k')_{k\in\mathbb{N}}$ is independent of $E$. Note that $\eta > 0$ since $\bar{\theta}_n$ is monotone decreasing in $n$. Now consider the following stopping time:

$$\tau' := \inf \left\{ n \geqslant 1 : \sum_{k=1}^{n} (\Psi_k' - \lambda\Delta) < -\eta \right\}. \tag{32}$$

It follows from (31) and (32) that $\mathbb{P}(\tau = \infty | E) \geqslant \mathbb{P}(\tau' = \infty)$. We next show that $\mathbb{P}(\tau' = \infty)$ is bounded away from 0.

Let $S_n := \sum_{k=1}^{n} (\Psi_k' - \lambda\Delta)$, with $S_0 := 0$. Since the $\Psi_k'$'s are i.i.d. with $\mathbb{E}\Psi_1' = \Delta$ and $|\Psi_k'| \leqslant 1$, it follows that $W_n := \exp(aS_n)$ is a Martingale w.r.t. $(\Psi_k')_{k\in\mathbb{N}}$, where '$a$' is the non-zero solution to $\mathbb{E}[\exp(a(\Psi_1' - \lambda\Delta))] = 1$ (Note that $\mathbb{E}\Psi_1' = \Delta > 0$ and $\lambda \in (0,1)$ ensures $a < 0$.). Fix an arbitrary $b > 0$ and define $T_{\eta,b} := \inf\{n \geqslant 1 : S_n \notin [-\eta, b]\}$ (We already know that $\eta > 0$.). By Doob's Optional Stopping Theorem [3], it follows that $\mathbb{E}W_{\min(T_{\eta,b},n)} = \mathbb{E}W_0 = 1$. Furthermore, since the stopped Martingale $W_{\min(T_{\eta,b},n)}$ is uniformly integrable, we in fact have $\mathbb{E}W_{T_{\eta,b}} = 1$. Thereafter using Markov's inequality, we obtain $\mathbb{P}\left( S_{T_{\eta,b}} < -\eta \right) = \mathbb{P}\left( W_{T_{\eta,b}} > e^{-\eta a} \right) \leqslant \exp(\eta a)$.

Since $b > 0$ is arbitrary, taking $\lim_{b \to \infty}$ on both sides and invoking the Bounded Convergence Theorem, we finally conclude that $\mathbb{P}(\tau' = \infty) = \mathbb{P}\left(S_{T_{\eta,\infty}} \geq -\eta\right) \geq 1 - \exp(\eta a)$, and hence

$$\mathbb{P}(\tau = \infty | E) \geq 1 - \exp(\eta a) > 0. \tag{33}$$

In conclusion,

$$\mathbb{P}\left(\bigcap_{m=1}^{\infty} \left| \sum_{j=1}^{m} \left(Y_{1,j}^{F_1} - Y_{2,j}^{F_2}\right) \right| \geq \theta_m \right) \geq \mathbb{P}(\tau = \infty) \geq \mathbb{P}(\tau = \infty | E)\mathbb{P}(E)$$

$$\underset{(*)}{\geq} (1 - \exp(\eta a)) \prod_{k=1}^{n_0} \mathbb{P}(\Psi_k > \bar{\theta}_k) > 0,$$

where $(*)$ follows from (30) and (33). Since $(F_1, F_2) \in \mathcal{G}(\mu_1) \times \mathcal{G}(\mu_2)$ is arbitrary, taking $\min_{F_1 \in \mathcal{G}(\mu_1), F_2 \in \mathcal{G}(\mu_2)}$ on both the sides above appealing to the fact that the $\mathcal{G}(\mu_i)$'s are finite, proves our assertion. $\square$

## D Proof of Theorem 3

Consider the first epoch and assign the labels $1, 2$ to the two arms picked to be played in this epoch. Suppose $N_i(n)$ denotes the number of times arm $i$ is played up to and including time $n$. Let $M_n := \min(N_1(n), N_2(n))$ and define the following stopping time:

$$\tau := \inf \left\{ n \geq 2 : \left| \sum_{k=1}^{M_n} (X_{1,k} - X_{2,k}) \right| < \theta_{M_n} \right\},$$

where the sequence $\Theta \equiv (\theta_m)_{m \in \mathbb{N}}$ is defined through (2) (main text). Then, $\tau$ denotes the time of the terminal play in the first epoch after which the algorithm starts over again. Recall that $t(i)$ denotes the type of arm $i$ and define the following conditional stopping times:

$$\tau_I := \tau \mid \{t(1) = t(2) = 2\}, \tag{34}$$
$$\tau_D := \tau \mid \{t(1) \neq t(2)\}, \tag{35}$$

where the subscripts $I$ and $D$ above indicate "Identical" and "Distinct" types, respectively. Let $S_n$ denote the cumulative pseudo-regret of UCB1 after $n$ plays in a stochastic two-armed bandit problem with separation $\Delta$. Recall that $R_n^\pi$ denotes the cumulative pseudo-regret of $\pi = \text{ALG}(\text{UCB1}, \Theta, 2)$ after $n$ plays; we shall suppress the superscript $\pi$ for notational simplicity and write $R_n$ for $R_n^\pi$. For any $n \in \mathbb{N}$, let $R_n'$ be an i.i.d. copy of $R_n$. Then, $R_n$ must satisfy the following stochastic recursive relation:

$$R_n = \mathbb{1}\{t(1) \neq t(2)\} S_{\min(\tau,n)} + \mathbb{1}\{t(1) = t(2) = 2\} \Delta \min(\tau, n) + R'_{n - \min(\tau, n)}$$

$$\leq \mathbb{1}\{t(1) \neq t(2)\} S_n + \mathbb{1}\{t(1) = t(2) = 2\} \Delta \tau + R'_{n - \min(\tau, n)}$$

$$= \mathbb{1}\{t(1) \neq t(2)\} S_n + \mathbb{1}\{t(1) = t(2) = 2\} \Delta \tau + \sum_{k=2}^{n} \mathbb{1}\{\tau = k\} R'_{n-k}$$

$$\leq \mathbb{1}\{t(1) \neq t(2)\} S_n + \mathbb{1}\{t(1) = t(2) = 2\} \Delta \tau + \mathbb{1}\{\tau \leq n\} R'_n, \tag{36}$$

where the last step holds since $R'_{n-k} \leq R'_n \; \forall \, k \leq n$ (this follows trivially since $\pi$ is agnostic to the length of the horizon of play[1]). Taking expectations on both sides of (36), we obtain

$$\mathbb{E}R_n \underset{(\dagger)}{\leq} 2\alpha(1-\alpha)\mathbb{E}S_n + (1-\alpha)^2 \Delta \mathbb{E}\tau_I + \left[2\alpha(1-\alpha)\mathbb{P}(\tau_D \leq n) + \alpha^2 + (1-\alpha)^2\right] \mathbb{E}R_n$$

$$\underset{(\ddagger)}{\leq} 2\alpha(1-\alpha)\mathbb{E}S_n + (1-\alpha)^2 \Delta \mathbb{E}\tau_I + \left[2\alpha(1-\alpha)(1-\beta) + \alpha^2 + (1-\alpha)^2\right] \mathbb{E}R_n$$

$$= 2\alpha(1-\alpha)\mathbb{E}S_n + (1-\alpha)^2 \Delta \mathbb{E}\tau_I + (1 - 2\beta\alpha(1-\alpha)) \mathbb{E}R_n$$

$$\implies \mathbb{E}R_n \leq \left(\frac{1}{\beta}\right) \mathbb{E}S_n + \left(\frac{(1-\alpha)\Delta \mathbb{E}\tau_I}{2\beta\alpha}\right),$$

where (†) uses (34), (35) and the fact that $\mathcal{D}(\mathcal{T}) = (\alpha, 1 - \alpha)$, and (‡) follows from part (i) of Lemma 2 (see Appendix F). We also know from part (ii) of Lemma 2 that $\mathbb{E}\tau_I < C_0$, where $C_0$ is a constant that depends on the user-defined parameters $(m_0, \gamma)$. The proof now concludes by invoking Theorem 1 of [1] for an upper bound on $\mathbb{E}S_n$ in order to obtain the desired upper bound on $\mathbb{E}R_n$, i.e.,

$$\mathbb{E}R_n \leqslant \left(\frac{8}{\beta\Delta}\right)\log n + \left(1 + \frac{\pi^2}{3} + \frac{(1-\alpha)C_0}{2\alpha}\right)\left(\frac{\Delta}{\beta}\right)$$

$$\leqslant 8\left(\beta\Delta\right)^{-1}\log n + \left(C_1 + \alpha^{-1}C_2\right)\beta^{-1}\Delta,$$

where $C_1 := 1 + \pi^2/3$ and $C_2 := C_0/2$. $\qquad\qquad\qquad\qquad\qquad\qquad\qquad\qquad\qquad\Box$

## E    Proof of Theorem 4

We begin by noting that the following is true for any integer $u > 1$ and $i \in \{1, 2\}$:

$$N_i(n) \leqslant u + \sum_{t=u+1}^{n} \mathbb{1}\{I_t = i,\ N_i(t) > u\},$$

where $I_t \in \{1, 2\}$ denotes the index of the arm played at time $t$. We set $u = (1/2 + \epsilon)n$ for an arbitrary $\epsilon \in (0, 1/2)$ and without loss of generality, carry out the rest of the analysis fixing $i = 1$. We have,

$$N_1(n) \leqslant \left(\frac{1}{2} + \epsilon\right)n + \sum_{t=\left(\frac{1}{2}+\epsilon\right)n+1}^{n} \mathbb{1}\left\{I_t = 1,\ N_1(t) > \left(\frac{1}{2} + \epsilon\right)n\right\}$$

$$\leqslant \left(\frac{1}{2} + \epsilon\right)n + \sum_{t=\left(\frac{1}{2}+\epsilon\right)n+1}^{n} \mathbb{1}\left\{I_t = 1,\ N_1(t) > \left(\frac{1}{2} + \epsilon\right)t\right\}$$

$$= \left(\frac{1}{2} + \epsilon\right)n + \sum_{t=\left(\frac{1}{2}+\epsilon\right)n+1}^{n} \mathbb{1}\left\{B_{1,t-1} > B_{2,t-1},\ N_1(t-1) > \left(\frac{1}{2} + \epsilon\right)t - 1\right\},$$

where $B_{i,t} := \overline{X}_i(t) + \sqrt{(2\log t)/N_i(t)}$ for $i \in \{1, 2\}$, with $\overline{X}_i(t)$ denoting the empirical mean reward from the first $N_i(t)$ plays of arm $i$. Therefore,

$$N_1(n) \leqslant \left(\frac{1}{2} + \epsilon\right)n + \sum_{t=\left(\frac{1}{2}+\epsilon\right)n}^{n-1} \mathbb{1}\left\{B_{1,t} > B_{2,t},\ N_1(t) \geqslant \left(\frac{1}{2} + \epsilon\right)t\right\}$$

$$= \left(\frac{1}{2} + \epsilon\right)n + Z_n, \qquad\qquad\qquad\qquad\qquad\qquad (37)$$

where $Z_n := \sum_{t=\left(\frac{1}{2}+\epsilon\right)n}^{n-1} \mathbb{1}\left\{B_{1,t} > B_{2,t},\ N_1(t) \geqslant \left(\frac{1}{2} + \epsilon\right)t\right\}$. Then,

$$\mathbb{E}Z_n$$

$$= \sum_{t=\left(\frac{1}{2}+\epsilon\right)n}^{n-1} \mathbb{P}\left(B_{1,t} > B_{2,t},\ N_1(t) \geqslant \left(\frac{1}{2} + \epsilon\right)t\right)$$

$$= \sum_{t=\left(\frac{1}{2}+\epsilon\right)n}^{n-1} \mathbb{P}\left(\frac{\sum_{j=1}^{N_1(t)} X_{1,j}}{N_1(t)} - \frac{\sum_{j=1}^{N_2(t)} X_{2,j}}{N_2(t)} > \sqrt{2\log t}\left(\frac{1}{\sqrt{N_2(t)}} - \frac{1}{\sqrt{N_1(t)}}\right),\ N_1(t) \geqslant \left(\frac{1}{2} + \epsilon\right)t\right)$$

$$= \sum_{t=\left(\frac{1}{2}+\epsilon\right)n}^{n-1} \mathbb{P}\left(\frac{\sum_{j=1}^{N_1(t)} Y_{1,j}}{N_1(t)} - \frac{\sum_{j=1}^{N_2(t)} Y_{2,j}}{N_2(t)} > \sqrt{2\log t}\left(\frac{1}{\sqrt{N_2(t)}} - \frac{1}{\sqrt{N_1(t)}}\right),\ N_1(t) \geqslant \left(\frac{1}{2} + \epsilon\right)t\right),$$

$$(38)$$

where $Y_{i,j} := X_{i,j} - \mathbb{E}X_{i,j}$ for $i \in \{1, 2\}$, $j \in \mathbb{N}$. Note that (38) follows since the mean rewards of both the arms are equal.

### E.1 Proof of part (i)

Consider an arbitrary non-negative integer $m \leqslant \left(\frac{1}{2} - \epsilon\right)t - 1$. Let $n_1(m) := \left(\frac{1}{2} + \epsilon\right)t + m$ and $n_2(m) := t - n_1(m)$. Then,

$$\mathbb{P}\left(\frac{\sum_{j=1}^{N_1(t)} Y_{1,j}}{N_1(t)} - \frac{\sum_{j=1}^{N_2(t)} Y_{2,j}}{N_2(t)} > \sqrt{2\log t}\left(\frac{1}{\sqrt{N_2(t)}} - \frac{1}{\sqrt{N_1(t)}}\right), \ N_1(t) = n_1(m)\right)$$

$$\leqslant \mathbb{P}\left(\frac{\sum_{j=1}^{n_1(m)} Y_{1,j}}{n_1(m)} - \frac{\sum_{j=1}^{n_2(m)} Y_{2,j}}{n_2(m)} > \sqrt{2\log t}\left(\frac{1}{\sqrt{n_2(m)}} - \frac{1}{\sqrt{n_1(m)}}\right)\right)$$

$$\underset{(\dagger)}{\leqslant} \exp\left(-4\left(\frac{t - 2\sqrt{n_1(m)n_2(m)}}{t}\right)\log t\right)$$

$$\underset{(\ddagger)}{\leqslant} \exp\left(-4\left(1 - \sqrt{1 - 4\epsilon^2}\right)\log t\right), \tag{39}$$

where ($\dagger$) follows using Hoeffding's inequality [4] and ($\ddagger$), since the product $n_1(m)n_2(m)$ is maximized on the set $\{m : 0 \leqslant m \leqslant (1/2 - \epsilon)t - 1\}$ at $m = 0$. From (38), we have

$$\mathbb{E}Z_n$$

$$= \sum_{t=\left(\frac{1}{2}+\epsilon\right)n}^{n-1} \sum_{m=0}^{\left(\frac{1}{2}-\epsilon\right)t-1} \mathbb{P}\left(\frac{\sum_{j=1}^{N_1(t)} Y_{1,j}}{N_1(t)} - \frac{\sum_{j=1}^{N_2(t)} Y_{2,j}}{N_2(t)} > \sqrt{2\log t}\left(\frac{1}{\sqrt{N_2(t)}} - \frac{1}{\sqrt{N_1(t)}}\right), \ N_1(t) = n_1(m)\right)$$

$$\underset{(\star)}{\leqslant} \sum_{t=\left(\frac{1}{2}+\epsilon\right)n}^{n-1} \sum_{m=0}^{\left(\frac{1}{2}-\epsilon\right)t-1} \exp\left(-4\left(1 - \sqrt{1 - 4\epsilon^2}\right)\log t\right)$$

$$= \left(\frac{1}{2} - \epsilon\right) \sum_{t=\left(\frac{1}{2}+\epsilon\right)n}^{n-1} t \exp\left(-4\left(1 - \sqrt{1 - 4\epsilon^2}\right)\log t\right)$$

$$\underset{(*)}{<} 2^{\rho(\epsilon)} n^{-(\rho(\epsilon)-1)}, \tag{40}$$

where ($\star$) follows from (39) and ($*$) holds with $\rho(\epsilon) := 3 - 4\sqrt{1 - 4\epsilon^2} > 0$ for $\epsilon > \sqrt{7}/8$. Now consider an arbitrary $\delta \in (0, 1)$. Then,

$$\mathbb{P}\left(\frac{N_1(n)}{n} \geqslant \left(\frac{1}{2} + \epsilon + \delta\right)\right) = \mathbb{P}\left(N_1(n) - \left(\frac{1}{2} + \epsilon\right)n \geqslant \delta n\right)$$

$$\leqslant \mathbb{P}(Z_n \geqslant \delta n) \qquad \text{(using (37))}$$

$$\leqslant \frac{\mathbb{E}Z_n}{\delta n} \qquad \text{(Markov's inequality)}$$

$$\leqslant \left(\frac{2^{\rho(\epsilon)}}{\delta}\right) n^{-\rho(\epsilon)} \qquad \text{(using (40))}$$

$$\leqslant \left(\frac{8}{\delta}\right) n^{-\rho(\epsilon)}.$$

Note that $\rho(\epsilon) \leqslant 0$ for $\epsilon \leqslant \sqrt{7}/8$. Thus, the above result trivially holds for all $\epsilon \in (0, 1/2)$. An identical result holds also for $N_2(n)$ by the symmetry of our proof. Therefore for any $i \in \{1, 2\}$, we have

$$\mathbb{P}\left(\left|\frac{N_i(n)}{n} - \frac{1}{2}\right| \geqslant \epsilon + \delta\right) \leqslant \left(\frac{8}{\delta}\right) n^{-\left(3 - 4\sqrt{1 - 4\epsilon^2}\right)}.$$

The form of the result stated in the theorem can be obtained by making the following substitutions order-wise: $\delta \leftarrow \delta'\epsilon'$, $\epsilon \leftarrow (1 - \delta')\epsilon'$, $\delta' \leftarrow \delta$, $\epsilon' \leftarrow \epsilon$. $\qquad\square$

## E.2 Proof of part (ii)

From (38), we have

$$
\begin{aligned}
&\mathbb{E}Z_n \\
&\leqslant \sum_{t=\left(\frac{1}{2}+\epsilon\right)n}^{n-1} \mathbb{P}\left(\frac{\sum_{j=1}^{N_1(t)} Y_{1,j}}{N_1(t)} - \frac{\sum_{j=1}^{N_2(t)} Y_{2,j}}{N_2(t)} > \sqrt{\frac{2\log t}{t}}\left(\frac{1}{\sqrt{\left(\frac{1}{2}-\epsilon\right)}} - \frac{1}{\sqrt{\left(\frac{1}{2}+\epsilon\right)}}\right),\ N_1(t) \geqslant \left(\frac{1}{2}+\epsilon\right)t\right) \\
&\leqslant \sum_{t=\left(\frac{1}{2}+\epsilon\right)n}^{n-1} \mathbb{P}\left(W_t > \frac{1}{\sqrt{\left(\frac{1}{2}-\epsilon\right)}} - \frac{1}{\sqrt{\left(\frac{1}{2}+\epsilon\right)}}\right),
\end{aligned}
\tag{41}
$$

where $W_t := \sqrt{\frac{t}{2\log t}}\left(\frac{\sum_{j=1}^{N_1(t)} Y_{1,j}}{N_1(t)} - \frac{\sum_{j=1}^{N_2(t)} Y_{2,j}}{N_2(t)}\right)$. Now,

$$
\begin{aligned}
&|W_t| \\
&\leqslant \sqrt{\frac{t}{2\log t}}\left(\left|\frac{\sum_{j=1}^{N_1(t)} Y_{1,j}}{N_1(t)}\right| + \left|\frac{\sum_{j=1}^{N_2(t)} Y_{2,j}}{N_2(t)}\right|\right) \\
&= \sqrt{\frac{t}{\log t}}\left(\sqrt{\frac{\log\log N_1(t)}{N_1(t)}}\left|\frac{\sum_{j=1}^{N_1(t)} Y_{1,j}}{\sqrt{2N_1(t)\log\log N_1(t)}}\right| + \sqrt{\frac{\log\log N_2(t)}{N_2(t)}}\left|\frac{\sum_{j=1}^{N_2(t)} Y_{2,j}}{\sqrt{2N_2(t)\log\log N_2(t)}}\right|\right) \\
&\leqslant \sqrt{\frac{t}{\log t}}\left(\sqrt{\frac{\log\log t}{N_1(t)}}\left|\frac{\sum_{j=1}^{N_1(t)} Y_{1,j}}{\sqrt{2N_1(t)\log\log N_1(t)}}\right| + \sqrt{\frac{\log\log t}{N_2(t)}}\left|\frac{\sum_{j=1}^{N_2(t)} Y_{2,j}}{\sqrt{2N_2(t)\log\log N_2(t)}}\right|\right) \\
&= \sqrt{\frac{\log\log t}{\log t}}\left(\sqrt{\frac{t}{N_1(t)}}\left|\frac{\sum_{j=1}^{N_1(t)} Y_{1,j}}{\sqrt{2N_1(t)\log\log N_1(t)}}\right| + \sqrt{\frac{t}{N_2(t)}}\left|\frac{\sum_{j=1}^{N_2(t)} Y_{2,j}}{\sqrt{2N_2(t)\log\log N_2(t)}}\right|\right).
\end{aligned}
\tag{42}
$$

Notice that the following can be deduced from part (i) of Theorem 4 using the Borel-Cantelli Lemma:

$$
\liminf_{t\to\infty} \frac{N_i(t)}{t} \geqslant \frac{1}{2} - \frac{\sqrt{3}}{4}\quad \text{w.p. } 1\ \forall\, i \in \{1,2\}.
\tag{43}
$$

In addition to the result in (43) that holds *w.p.* 1, we also know that $N_i(t)$, for any $i \in \{1,2\}$ and $t \geqslant 0$, can be lower bounded *pathwise* by a deterministic non-decreasing function of time, say $\lambda(t)$, that grows to $+\infty$ as $t \to \infty$. This is a trivial consequence due to the structure of the UCB1 policy and the fact that the rewards are bounded. We therefore have for any $i \in \{1,2\}$,

$$
\left|\frac{\sum_{j=1}^{N_i(t)} Y_{i,j}}{\sqrt{2N_i(t)\log\log N_i(t)}}\right| \leq \sup_{m \geq \lambda(t)}\left|\frac{\sum_{j=1}^{m} Y_{i,j}}{\sqrt{2m\log\log m}}\right|.
$$

Now for any fixed $i \in \{1,2\}$, $\mathbb{E}Y_{i,j} \sim$ i.i.d. $\forall\, j$ with $\mathbb{E}Y_{i,1} = 0$ and $\text{Var}\,(Y_{i,1}) = \text{Var}\,(X_{i,1}) \leqslant 1$. Also, $\lambda(t)$ is non-decreasing and $\lambda(t) \uparrow \infty$. Therefore, the Law of the Iterated Logarithm [5] implies

$$
\limsup_{t\to\infty}\left|\frac{\sum_{j=1}^{N_i(t)} Y_{i,j}}{\sqrt{2N_i(t)\log\log N_i(t)}}\right| \leqslant 1\quad \text{w.p. } 1\ \forall\, i \in \{1,2\}.
\tag{44}
$$

From (42), (43) and (44), we conclude that

$$
\lim_{t\to\infty} W_t = 0\quad \text{w.p. } 1.
\tag{45}
$$

Now consider an arbitrary $\delta > 0$. Then,

$$
\mathbb{P}\left(\frac{N_1(n)}{n} \geqslant \left(\frac{1}{2} + \epsilon + \delta\right)\right) = \mathbb{P}\left(N_1(n) - \left(\frac{1}{2} + \epsilon\right)n \geqslant \delta n\right)
$$

$$
\underset{(\dagger)}{\leqslant} \mathbb{P}(Z_n \geqslant \delta n)
$$

$$
\underset{(\ddagger)}{\leqslant} \frac{\mathbb{E}Z_n}{\delta n}
$$

$$
\underset{(\star)}{\leqslant} \frac{1}{\delta n} \sum_{t=\left(\frac{1}{2}+\epsilon\right)n}^{n-1} \mathbb{P}\left(W_t > \frac{1}{\sqrt{\left(\frac{1}{2}-\epsilon\right)}} - \frac{1}{\sqrt{\left(\frac{1}{2}+\epsilon\right)}}\right),
$$

where ($\dagger$) follows using (37), ($\ddagger$) using Markov's inequality and ($\star$) from (41). Now,

$$
\mathbb{P}\left(\frac{N_1(n)}{n} \geqslant \left(\frac{1}{2} + \epsilon + \delta\right)\right) \leqslant \frac{1}{\delta n} \sum_{t=\left(\frac{1}{2}+\epsilon\right)n}^{n-1} \mathbb{P}\left(W_t > \frac{1}{\sqrt{\left(\frac{1}{2}-\epsilon\right)}} - \frac{1}{\sqrt{\left(\frac{1}{2}+\epsilon\right)}}\right)
$$

$$
\leqslant \left(\frac{\frac{1}{2}-\epsilon}{\delta}\right) \sup_{\left(\frac{1}{2}+\epsilon\right)n \leqslant t \leqslant n-1} \mathbb{P}\left(W_t > \frac{1}{\sqrt{\left(\frac{1}{2}-\epsilon\right)}} - \frac{1}{\sqrt{\left(\frac{1}{2}+\epsilon\right)}}\right)
$$

$$
\leqslant \left(\frac{\frac{1}{2}-\epsilon}{\delta}\right) \sup_{t \geqslant n/2} \mathbb{P}\left(W_t > \frac{1}{\sqrt{\left(\frac{1}{2}-\epsilon\right)}} - \frac{1}{\sqrt{\left(\frac{1}{2}+\epsilon\right)}}\right). \qquad (46)
$$

Using (45) and (46), we conclude that

$$
\limsup_{n\to\infty} \mathbb{P}\left(\frac{N_1(n)}{n} \geqslant \left(\frac{1}{2} + \epsilon + \delta\right)\right) \leqslant \left(\frac{\frac{1}{2}-\epsilon}{\delta}\right) \limsup_{n\to\infty} \mathbb{P}\left(W_n > \frac{1}{\sqrt{\left(\frac{1}{2}-\epsilon\right)}} - \frac{1}{\sqrt{\left(\frac{1}{2}+\epsilon\right)}}\right) = 0.
$$

Since $\delta > 0$ is arbitrary, it follows that $\lim_{n\to\infty} \mathbb{P}\left(\frac{N_1(n)}{n} \geqslant \frac{1}{2} + \epsilon\right) = 0$ for any $\epsilon > 0$. Since our proof is symmetric w.r.t. the arms, we also have $\lim_{n\to\infty} \mathbb{P}\left(\frac{N_2(n)}{n} \geqslant \frac{1}{2} + \epsilon\right) = 0 \implies \lim_{n\to\infty} \mathbb{P}\left(\frac{N_1(n)}{n} \leqslant \frac{1}{2} - \epsilon\right) = 0$. Therefore, $\lim_{n\to\infty} \mathbb{P}\left(\left|\frac{N_i(n)}{n} - \frac{1}{2}\right| \geqslant \epsilon\right) = 0$ for $i \in \{1, 2\}$ and any $\epsilon > 0$. $\qquad\square$

## F   Ancillary results

**Lemma 1** *Consider a stochastic two-armed bandit with rewards bounded in $[0, 1]$. Suppose that the reward distributions of the two arms $(F_1, F_2) \in \mathcal{G}(\mu_1) \times \mathcal{G}(\mu_2)$ satisfy Assumption 1 (main text). Let $N_i(n)$ denote the number of times arm $i$ is played by UCB1 [1] up to and including time $n$. At any time $n^+$, $(X_{i,k})_{k=1}^m$ denotes the sequence of rewards realized from the first $m \leqslant N_i(n)$ plays of arm $i$. For each $n \in \mathbb{N}$, let $M_n := \min\left(N_1(n), N_2(n)\right)$ and consider the following stopping times:*

$$
\tau := \inf\left\{n \geqslant 2 : \left|\sum_{k=1}^{M_n} (X_{1,k} - X_{2,k})\right| < \theta_{M_n}\right\}, \qquad (47)
$$

$$
\tau' := \inf\left\{n \geqslant 1 : \left|\sum_{k=1}^{n} (X_{1,k} - X_{2,k})\right| < \theta_n\right\}, \qquad (48)
$$

*where the sequence $\Theta \equiv \{\theta_n : n = 1, 2, ...\}$ is defined through (2) (main text). Then, $M_\tau = \tau'$ pathwise.*

**Lemma 2** *Consider the setting of Lemma 1. Recall that $\mathcal{T} = \{1, 2\}$ and $t(i) \in \mathcal{T}$ denotes the type of arm $i$. Define the following conditional stopping times:*

$$
\tau_D := \tau \mid t(1) \neq t(2), \qquad (49)
$$

$$
\tau_I := \tau \mid t(1) = t(2), \qquad (50)
$$

*where the subscripts D and I indicate "Distinct" and "Identical" types, respectively. Then, the following results hold:*

(i) $\mathbb{P}(\tau_D = \infty) \geqslant \beta$, *where $\beta$ is as defined in* (1) *(main text).*

(ii) $\mathbb{E}\tau_I < C_0$, *where $C_0$ is a constant that depends on the user-defined parameters $(m_0, \gamma)$ featuring in* (2) *(main text) that ensure $\Theta$ satisfies the conditions of Proposition 1 (main text).*

### F.1 Proof of Lemma 1

We begin by noting the following facts:

1. *Fact 1:* $(M_n)_{n \geqslant 2}$ is a non-decreasing sequence of natural numbers (starting from $M_2 = 1$), with $M_{n+1} \leqslant M_n + 1$.

2. *Fact 2:* For each $i \in \{1, 2\}$, $\liminf_{n \to \infty} N_i(n) = \infty$ pathwise[2] (consequence of UCB1 and bounded rewards). Consequently, $\liminf_{n \to \infty} M_n = \infty$ pathwise.

Define $\Psi_k := X_{1,k} - X_{2,k}$. Fix some $m \in \mathbb{N}$ and consider an arbitrary sample-path $\omega$ such that $M_\tau(\omega) = m$. Then on $\omega$, we must also have $m = \inf \left\{ l \geqslant 1 : \left| \sum_{k=1}^{l} \Psi_k(\omega) \right| < \theta_l \right\}$ (follows from the definitions of $\tau$ and $\tau'$). Since the choice of $m$ is arbitrary (due to *Fact 1* and *Fact 2*), it must be that on any arbitrary $\omega$, $M_\tau(\omega) = \inf \left\{ l \geqslant 1 : \left| \sum_{k=1}^{l} \Psi_k(\omega) \right| < \theta_l \right\}$. The assertion thus follows. □

### F.2 Proof of Lemma 2 part (i)

We know from Lemma 1 that $M_\tau = \tau'$. In particular, this also implies $M_{\tau_D} = \tau' \mid t(1) \neq t(2)$. Notice that $\tau_D \geqslant 2M_{\tau_D}$ is always true. Thus, it follows that $\tau_D \geqslant 2\tau' \mid t(1) \neq t(2)$. Therefore, $\mathbb{P}(\tau_D = \infty) \geqslant \mathbb{P}(\tau' = \infty \mid t(1) \neq t(2)) = \mathbb{P}(\tau' = \infty \mid t(1) = 1, t(2) = 2) \geqslant \beta$ (Recall from (1) (main text) the definition of $\beta$.). The assertion thus follows. □

### F.3 Proof of Lemma 2 part (ii)

Throughout this proof, the condition $t(1) = t(2)$ is implicit and we shall avoid writing it explicitly to simplify notation. Let $\Psi_k := X_{1,k} - X_{2,k}$. Consider the following:

$$
\begin{aligned}
\mathbb{P}(\tau_I > n) &= \mathbb{P}\left( \bigcap_{l=2}^{n} \left\{ \left| \sum_{k=1}^{M_l} \Psi_k \right| \geqslant \theta_{M_l} \right\} \right) \\
&\leqslant \mathbb{P}\left( \left| \sum_{k=1}^{M_n} \Psi_k \right| \geqslant \theta_{M_n} \right) \\
&= \sum_{m=1}^{n} \mathbb{P}\left( \left| \sum_{k=1}^{M_n} \Psi_k \right| \geqslant \theta_{M_n}, \ N_1(n) = m \right) \\
&= \sum_{m=1}^{n} \mathbb{P}\left( \left| \sum_{k=1}^{\min(m,n-m)} \Psi_k \right| \geqslant \theta_{\min(m,n-m)}, \ N_1(n) = m \right).
\end{aligned}
$$

Consider an arbitrary $\kappa \in \left(0, 1/2 - \sqrt{3}/4\right)$. Splitting the above summation three-ways, we obtain

$$\mathbb{P}(\tau_I > n) \leqslant \sum_{m=1}^{\kappa n} \mathbb{P}\left(N_1(n) = m\right) + \sum_{m=\kappa n}^{(1-\kappa)n} \mathbb{P}\left(\left|\sum_{k=1}^{\min(m,n-m)} \Psi_k\right| \geqslant \theta_{\min(m,n-m)}\right)$$

$$+ \sum_{m=(1-\kappa)n}^{n} \mathbb{P}\left(N_1(n) = m\right)$$

$$\leqslant \mathbb{P}\left(N_1(n) \leqslant \kappa n\right) + \mathbb{P}\left(N_2(n) \leqslant \kappa n\right) + \sum_{m=\kappa n}^{(1-\kappa)n} \mathbb{P}\left(\left|\sum_{k=1}^{\min(m,n-m)} \Psi_k\right| \geqslant \theta_{\min(m,n-m)}\right)$$

$$\leqslant \mathbb{P}\left(N_1(n) \leqslant \kappa n\right) + \mathbb{P}\left(N_2(n) \leqslant \kappa n\right) + 2\sum_{m=\kappa n}^{(1-\kappa)n} \exp\left(\frac{-\theta_{\min(m,n-m)}^2}{2\min(m, n-m)}\right),$$

where the last step follows from Hoeffding's inequality [4] using the fact that $\Psi_k$'s are i.i.d. with $\mathbb{E}\Psi_1 = 0$ and $|\Psi_1| \leqslant 1$. Recall that for any $\kappa \in \left(0, 1/2 - \sqrt{3}/4\right)$, part (i) of Theorem 4 guarantees that $\sum_{n=1}^{T}\left(\mathbb{P}\left(N_1(n) \leqslant \kappa n\right) + \mathbb{P}\left(N_2(n) \leqslant \kappa n\right)\right) = \mathcal{O}_T(1)$ (the subscript $T$ is added to indicate that the asymptotic scaling is w.r.t. $T$), with the limit being a constant that depends on the user-defined parameters $(m_0, \gamma)$ determining the sequence $(\theta_m)_{m \in \mathbb{N}}$ in (2) (main text). Therefore, we have

$$\sum_{n=1}^{T} \mathbb{P}(\tau_I > n) \leqslant \mathcal{O}_T(1) + 2\sum_{n=1}^{T}\sum_{m=\kappa n}^{(1-\kappa)n} \exp\left(\frac{-\theta_{\min(m,n-m)}^2}{2\min(m, n-m)}\right). \tag{51}$$

To analyze the double-summation term, consider the following:

$$\sum_{m=\kappa n}^{(1-\kappa)n} \exp\left(\frac{-\theta_{\min(m,n-m)}^2}{2\min(m, n-m)}\right) \leqslant \sum_{m=\kappa n}^{n/2} \exp\left(\frac{-\theta_m^2}{2m}\right) + \sum_{m=n/2}^{(1-\kappa)n} \exp\left(\frac{-\theta_{n-m}^2}{2(n-m)}\right)$$

$$\leqslant 2\sum_{m=\kappa n}^{\infty} \exp\left(\frac{-\theta_m^2}{2m}\right)$$

$$\leqslant 2\sum_{m=\kappa n}^{\infty} \exp\left(\frac{-\theta_{m-m_0}^2}{2(m-m_0)}\right), \tag{52}$$

Notice that

$$\frac{\theta_{m-m_0}^2}{2(m-m_0)} = \left(1 - \frac{m_0}{m}\right)(2\log m + (\gamma/2)\log\log m) = 2\log m + (\gamma/2)\log\log m + o_m(1), \tag{53}$$

where the last equality follows since $m_0$ and $\gamma$ are finite user-defined parameters. Using (52) and (53), we obtain

$$\sum_{m=\kappa n}^{(1-\kappa)n} \exp\left(\frac{-\theta_{\min(m,n-m)}^2}{2\min(m, n-m)}\right) \leqslant 2\sum_{m=\kappa n}^{\infty} \exp\left(-\left(2\log m + (\gamma/2)\log\log m + o_m(1)\right)\right)$$

$$= 2\sum_{m=\kappa n}^{\infty} \frac{\mathcal{O}_m(1)}{m^2 (\log m)^{\gamma/2}}$$

$$\leqslant \frac{1}{(\log n + \log \kappa)^{\gamma/2}} \sum_{m=\kappa n}^{\infty} \frac{\mathcal{O}_m(1)}{m^2}$$

$$= \mathcal{O}_n\left(\frac{1}{(\log n + \log \kappa)^{\gamma/2}}\left(\frac{1}{\kappa n} + \frac{1}{\kappa^2 n^2}\right)\right). \tag{54}$$

From (51) and (54), it follows that

$$\sum_{n=1}^{T} \mathbb{P}(\tau_I > n) \leqslant \mathcal{O}_T(1) + \sum_{n=1}^{T} \mathcal{O}_n\left(\frac{1}{(\log n + \log \kappa)^{\gamma/2}}\left(\frac{1}{\kappa n} + \frac{1}{\kappa^2 n^2}\right)\right)$$

$$= \mathcal{O}_T(1),$$

where the conclusion in the last step follows since $\gamma > 2$ is a finite user-defined parameter and $\kappa \in \left(0, 1/2 - \sqrt{3}/4\right)$ is arbitrarily chosen. Therefore, the stated assertion that $\mathbb{E}\tau_I < C_0$, where $C_0$ is some finite constant that depends on $(m_0, \gamma)$, follows. $\qquad \square$

**Remark.** Part (i) of Theorem 4 has a significant bearing on this result. Specifically, if unlike UCB1, the playing rule does not satisfy a concentration property akin to the one stated in part (i) of Theorem 4, then the $\mathcal{O}_T(1)$ term on the RHS in (51) would instead be $\Omega(T)$.

# G  The CAB problem with $|\mathcal{T}| = K$

In this section, we extend our results to $K$-typed settings. Let $\mathcal{T} = \{1, 2, ..., K\}$ and $\mathcal{D}(\mathcal{T})$ denote the distribution over $\mathcal{T}$. The mean reward associated with type $i \in \mathcal{T}$ is denoted by $\mu_i$. We assume that the mean rewards associated with each of the $K$ types are *distinct*[3], and without loss of generality, assume that type 1 is optimal, i.e., $\mu_1 > \mu_i \ \forall \ i \in \mathcal{T}\backslash\{1\}$. The sub-optimality gap of type $i$ is denoted by $\Delta_i := \mu_1 - \mu_i$, and the minimal separation between any pair of types, by $\Delta_0 := \min_{(i,j)\in\mathcal{T}^2:i\neq j} |\mu_i - \mu_j|$. In § G.1 and § G.2, we propose gap-aware and gap-agnostic algorithms for the CAB problem with $K$ types and state their performance guarantees (without proof).

## G.1  A near-optimal gap-aware algorithm

Below, we present a simple fixed-design ETC (Explore-then-Commit) algorithm assuming ex ante knowledge of the duration of play[4] $n$ and a separability parameter $\delta \in (0, \Delta_0]$. It is noteworthy that the informational requirement is significantly greater in the CAB problem compared to its finite-armed counterpart as it assumes knowledge of a lower bound on the minimal separation between any pair of types ($\Delta_0$), instead of the minimal sub-optimality gap ($\min_{i>1} \Delta_i$) which is relatively coarser information ($\because \min_{i>1} \Delta_i > \Delta_0$).

---

**Algorithm 1** ETC-$\infty$(K): ETC for an infinite population of arms with $|\mathcal{T}| = K$.

---

1: **Input:** $(n, \delta)$, where $\delta \in (0, \Delta_0]$.
2: Set epoch length $L = \lceil 2\delta^{-2} \log n \rceil$. Set budget $T = n$.
3: **Initialization:** Select two *new* arms. Call it consideration set $\mathcal{A} = [K]$.
4: $m \leftarrow \min\left(L, T/K\right)$.
5: Play each arm in $\mathcal{A}$ $m$ times. Update budget: $T \leftarrow T - Km$.
6: **if** $\left|\sum_{k=1}^{m}(X_{i,k} - X_{j,k})\right| < \delta m$ for any distinct pair $(i,j) \in \mathcal{A}^2$ **then**
7: $\quad$ Permanently discard $\mathcal{A}$ and go to **Initialization**.
8: **else**
9: $\quad$ Commit the remaining budget of play to arm $i^* \in \arg\max_{i \in \mathcal{A}} \sum_{k=1}^{m} X_{i,k}$.

---

The stated version of the algorithm does not generalize well to $K$ types. In order to appreciate this, consider the particular case of a uniform distribution over $\mathcal{T}$. In this case, a new arm is equally likely to be any one of the $K$ possible types and therefore, it would take the algorithm $K^K$ fresh draws in expectation of size $K$ consideration sets in order to obtain one that is fully heterogeneous (contains one arm of each type). Thus, the expected cumulative regret would grow proportionally to $K^K$ which is unacceptable. This can be improved to an $\mathcal{O}(K \log K)$ dependence on the number of types by suitable tweaks of the algorithm. Specifically, a natural modification would be to start with a consideration set containing a single arm and augmenting it sequentially by adding new arms (one at a time) that are sufficiently separated from each arm in the set. Instructively, the stopping point would occur when the algorithm accumulates $K$ arms that are all different from each other, after which it would simply commit the residual sampling budget to the empirically best arm. One can show that the improvement due to said modification is significant in the sense that the regret of the modified algorithm scales as $\mathcal{O}\left((\log K)\left(\sum_{i=2}^{K}\Delta_i\right)\delta^{-2}\log n\right)$. However, the analysis of the modified algorithm, albeit similar in spirit to the regret analysis of Algorithm 1 (main text), is quite tedious and becomes further so if one were to consider generic $K$-point distributions over $\mathcal{T}$ (instead of the uniform distribution) and is therefore omitted from this text.

## G.2 A near-optimal gap-agnostic algorithm

Below, we present a generalization of our framework for the CAB problem with a binary $\mathcal{T}$ (ALG($\pi, \Theta, 2$), main text), adapted to an arbitrary finite cardinality $\mathcal{T}$.

---

**Algorithm 2** ALG($\Xi, \Theta, K$): An algorithmic framework for countable-armed bandits with $|\mathcal{T}| = 2$.

---
1: **Input:** A $\Delta$-agnostic playing rule $\Xi$, a deterministic sequence $\Theta \equiv \{\theta_m : m = 1, 2, ...\}$ in $\mathbb{R}$.
2: **Initialization (Starts a new epoch):** Select $K$ *new* arms. Call it consideration set $\mathcal{A} = [K]$.
3: For $s \in [K]$, play each arm in $\mathcal{A}$ once.
4: $m \leftarrow 1$.
5: **for** $s \in \{K+1, K+2, ...\}$ **do**
6:     **if** $|\sum_{k=1}^{m} (X_{i,k} - X_{j,k})| < \theta_m$ for any distinct pair $(i,j) \in \mathcal{A}^2$ **then**
7:         Permanently discard $\mathcal{A}$ and go to **Initialization**.
8:     **else**
9:         Play an arm from $\mathcal{A}$ according to $\Xi$.
10:         $m \leftarrow \min_{i \in \mathcal{A}} N_i(s)$.

---

**Proposition 1 (Lower bound on the true negative rate)** *For each $i \in \mathcal{T} = [K]$, let $\left(Y_{i,k}^{F_i}\right)_{k \in \mathbb{N}}$ denote an i.i.d. sequence of random variables with distribution $F_i \in \mathcal{G}(\mu_i)$ satisfying Assumption 1 (main text). Let $\Theta \equiv \{\theta_m : m = 1, 2, ...\}$ be a deterministic non-negative real-valued sequence such that $\{(\theta_m/m) : m = 1, 2, ...\}$ is monotone decreasing in $m$ with $\theta_1 < 1$ and $\theta_m = o(m)$. For each $(i,j) \in [K]^2$ s.t. $i < j$, define the following stopping time:*

$$\tau_{i,j} := \inf \left\{ n \in \mathbb{N} : \left| \sum_{k=1}^{n} \left(Y_{i,k}^{F_i} - Y_{j,k}^{F_j}\right) \right| < \theta_n \right\}.$$

*Then,*

$$\widetilde{\beta} := \min_{F_1 \in \mathcal{G}(\mu_1),...,F_K \in \mathcal{G}(\mu_K)} \mathbb{P}\left( \min_{(i,j) \in [K]^2 : i<j} \tau_{i,j} = \infty \right) > 0. \tag{55}$$

**Remark.** If $K = 2$, then $\widetilde{\beta} = \beta$, where $\beta$ is as defined in (1) (main text).

**Proposition 2 (Upper bound on the expected regret of ALG(UCB1, $\Theta, K$))** *Consider the input sequence $\Theta \equiv \{\theta_m : m = 1, 2, ...\}$ given by*

$$\theta_m := \sqrt{m^2(m+m_0)^{-1} \left(4\log(m+m_0) + \gamma \log\log(m+m_0)\right)}, \tag{56}$$

*where $m_0 \geqslant 0$ and $\gamma > 2$ are user-defined parameters that ensure $\Theta$ satisfies the conditions of Proposition 1 (for example, $m_0 = 11$ and $\gamma = 2.1$ is an acceptable configuration). Suppose that Assumption 1 (main text) is satisfied. Then, the expected cumulative regret of $\pi = ALG(UCB1, \Theta, K)$ after any number of plays $n$ is bounded as follows:*

$$\mathbb{E}R_n^\pi \leqslant \min \left[ \left( \max_{i \in \{2,...,K\}} \Delta_i \right) n, \, 8\widetilde{\beta}^{-1} \left( \sum_{i=2}^{K} \Delta_i^{-1} \right) \log n + \left(C_1 + \alpha^{-1}C_2\right) \widetilde{\beta}^{-1} \left( \sum_{i=2}^{K} \Delta_i \right) \right], \tag{57}$$

*where $\widetilde{\beta}$ is as defined in (55), $\Delta_i = \mu_1 - \mu_i > 0$ for $i \in \{2, ..., K\}$, $C_1$ is an absolute constant and $C_2$ is a constant that depends only on the free parameters of the algorithm, namely $(m_0, \gamma)$.*

**Remark.** The constants $C_1, C_2$ above are the same as the ones appearing in Theorem 3 (main text).

## Footnotes

[1]We could not claim this directly for Algorithm 1 as it depended on ex ante knowledge of the length of play.

[2]For unbounded rewards, this would hold w.p. 1, not pathwise.

[3]This is a critical assumption, which if violated will cause our algorithms to incur linear regret.

[4]The standard exponential doubling trick can be employed to make the algorithm horizon-free, cf. [2].