[Reviews · NeurIPS 2020]

Review 1

Summary and Contributions: This paper proposes a new bandit model at the intersection of the classical MAB and the infinitely armed bandit. Infinite reservoirs of arms are gathered in K categories, one of them being the optimal one. The learner must find the right category, that is the one more likely to contain the/an optimal arm. TL;DR: I think it is a neat idea and despite some minor questions and remarks, I recommend to accept this paper.

Strengths: * Novelty: this is a completely new model, with a new range of possible applications. * Theoretical grounding: It also comes with new technical results of independent interest. * The paper is well written overall

Weaknesses: * Clarity: Amibuity between "Play" (an arm, i.e. sample a *reward* from a fixed distribution) and "pick" (an arm, i.e. sample a new distribution from a category of arms, not a reward). I think this should be made super clear in your setting defition. Choose two words and clearly define what they mean in maths terms so that the reader does loose track.

Correctness: The paper and the appendix are very clear and I could not spot any major mistake but I may have missed something.

Clarity: Yes.

Relation to Prior Work: Good overall. I think you missed this one: Jedor, Matthieu, Vianney Perchet, and Jonathan Louedec. "Categorized Bandits." Advances in Neural Information Processing Systems. 2019. And maybe a few related to this one (see refs therein).

Reproducibility: Yes

Additional Feedback: UPDATE: Thank you for your answers, great work, I raised my score . Main remarks: * Algorithm clarity: related to the ambiguity problem raised above, I think your algorithm 1 is a little bit hard to parse. Could you please clarify when you pick a new arm from a category and when you actually receive a reward ? Minor remarks: * l.71: extra ":" before whether


Review 2

Summary and Contributions: The paper studies a novel bandit setting with countably many arms sampled from a finite set of types. The paper provides algorithms with lower and upper regret bounds.

Strengths: ++ Afaik only a handful of papers have studied the countably-armed bandit problem so far, despite its practical relevance. Even more so, the submission seems to be the first paper considering the restriction to finitely-many types, resulting (potentially) improved bounds.

Weaknesses: -- The bandit algorithms are developed to support the proofs, and they are rather clumsy and wasteful and likely would not perform well in practice. -- The β in Eq.(1) depends on Q1 and Q2 and hence on Δ, and β→0 *somehow* for Δ→0. Therefore the main bound 𝔼R≤8log(n)/βΔ+... does not scale like 1/Δ as for 2-armed bandits, but worse, as the authors readily admit in Line 252. Unfortunately they do not discuss *how* β scales with Δ. This is quite disappointing, because now it is unclear whether the bound is any better than the ones in papers that do not assume finitely-many types. [authors respond that empirically β~Δ, which is promising] - Strong/unrealistic assumptions make this work more restrictive than previous work: o Restriction to finitely many types, o The support of every type has to include 0 and 1, o No two arms are allowed to have the same expected reward (no zero gaps). - The algorithm depends on knowing (an upper bound on) the smallest gap between any two arms Δ0. - The bound deteriorates not w.r.t. to the usual gap between best and second-best arm, but with Δ0, which for large K would typically be much smaller.

Correctness: + The claims and informal proof ideas all seem plausible and correct, though I haven't checked the formal proofs in the supplementary material.

Clarity: ++ The paper has been "optimized" for clarity: It focusses on the simplest 2-type and Δ-aware case first, then the Δ-agnostic case, with results accompanied by intuition why they are true and proof ideas, while the more interesting countable-type case and the formal proofs are delegated to the supplementary.

Relation to Prior Work: The closest work to compare to would be the finite-armed bandit problem, and the countable-armed bandit problem with a rich (uncountable) class of types. The former comparison could be improved by stating the Lai&Robbins lower bound explicitly. Some of the latter papers are cited, but a more quantitative comparison is desirable. If I remember correctly: (a) all these works make some scaling assumption close to the maximum of the reward domain [correction by author: scaling of distribution of mean rewards.], while the authors assume 'support at 0&1' and 'no two means are the same'. (b) none of these algorithms is 'optimal' for all n, and maybe no algorithm can be. Optimizing for finite n seems to necessarily deteriorate asymptotic performance. In contrast, the authors' work *seems* to not have this problem. Which assumption is responsible for this? A discussion of (a) and (b) in comparison to the authors' assumptions would be good, and given the dependence of β on Δ (see above) makes it imo even necessary. The following 3 further countable-arm bandit papers are probably worth to discuss briefly: Alexandra Carpentier and Michal Valko (ICML 2015) Simple regret for infinitely many armed bandits http://www.jmlr.org/proceedings/papers/v37/carpentier15.pdf Apart from other things, they estimate parameter beta (online). Y David, N Shimkin (2014) Infinitely many-armed bandits with unknown value distribution https://link.springer.com/chapter/10.1007/978-3-662-44848-9_20 The assume arms are deterministic, but "mean" reward for new arm sampled has *unknown* i.i.d. distribution. Arghya Roy Chaudhuri and Shivaram Kalyanakrishnan (UAI 2018) Quantile-Regret Minimisation in Infinitely Many-Armed Bandits Their algorithm (time-doubling loop over MOSS) for unknown reward distribution, primarily considers a weaker quantile regret notion, but can be converted to sub-optimal classical regret.

Reproducibility: Yes

Additional Feedback: - Please discuss the gap between the lower bound factor α(1-α) in Line 140, which is missing in the upper bound Eq(3), esp. in the limit α→0|1. [gap has been closed meanwhile by the authors] - Alg.1 picks an arm, labels it 1, and samples it m times once. This (X_{1,k}) never gets updated. Only Label 2 gets updated. So Alg.1 is stuck with a potentially bad sample. [This implies that the algorithm depends on and is only ‘good’ for this fixed n] Alg.2 discards all K arms [authors say yes] and the text "(2m+1)th play" hints at that this may not be intended [still a bit confused]. - In Thm.2, the 1/α can be quite big and o(1) is only asymptotic. Since Thm.2 is only a warm-up for Thm.3 this may not be worth a comment, but if the authors have some ready one-line answers, maybe worth to share. - I find Section 4 the most interesting: The super-slow convergence of UCB1 and the failure of Thompson sampling. Could both be caused by the/a same underlying deeper reason? - Can you explain why the argument in Lines 269-272 implies a multiplicatively worse bound (1/β), rather than an extra additive loss? - There is a (potential) mismatch between the motivating examples in lines 29-38 and the proven results: Identifying arms with users, where users come in K(=2) types is weird: If there were a single possible Ad, the bandit algorithm would try to choose an arm=user of type most interested in this Ad, but in practice ad-algorithms work the other way round: A user comes, and the algorithm has to choose an Ad. In the provided examples K seems to rather correspond to some sort of a side information, not to types, which the algorithm has no control over. For instance, the algorithm should not be allowed to send away patients (ask for new arms in your model), just because they are not of the right type and treat a single patient who can be cured. Maybe a suitable example could be that you have plenty of ads which can be clustered such that their payoff is almost the same, and you have to decide which ads to show to the next user. But this is an approximate model (i.e. Ads are never exactly the same), and it would need side information to make it practical. Or choose the best Ad that fits all the users on average might also work. Minor comments: - Define Filtration F_n explicitly - Comment on that pseudo-regret is weaker than regret - Line 123: 'does not depend on any problem primitive' does that mean it is a concrete and fixed real number, which just has not been worked out, or could it depend on 'non-problem primitives' whatever that may be, e.g. free parameters of the algorithm? - Line 101 Q∈[0;1] implies and is stronger than Line 136 μ(Q)∈[0;1]. I presume Thm.1 assumes the former not just the latter. If so, say so. - Line 139: Why the restriction to asymptotically consistent π? Aren't asymptotically non-consistent π even worse? - Comment that Thm.4(ii) does not follow from (i). - Suppl.Line 248: ALG(UCB1,θ,2) -> ALG(UCB1,θ,K)


Review 3

Summary and Contributions: UPDATE AFTER SEEING THE REBUTTAL In light of the issue about beta, which can mask the true dependence on Delta (the author's mention it can introduce a 1/Delta^2 dependence in some regime, and in other regimes it still is unclear how beta behaves), I am slightly reducing my score. However, I am still overall positive about this work, especially due to the anti-concentration results for UCB. ------------------------------------------------ This work explores what the authors dub the countable-armed bandit problem: a stochastic multi-armed bandit problem where there there are infinitely many arms but each arm has a type from a finite number of types (with all arms of the same type having the same reward distribution). The authors explore instance-dependent regret bounds in this setting. For simplicity, the main text is restricted to the case of two types. The authors first give a lower bound for this problem (which, surprisingly, appears to be inferior to lower bounds for a basic, two-armed bandit problem). Then, as a warm-up step, they develop an explore-then-commit style algorithm which uses information about the gap Delta, and they show that this algorithm has a good regret bound. They then get to their main algorithm: a master/slave type algorithm that takes as input some standard stochastic multi-armed bandit algorithm (like UCB) pi and which does not require information about the gap Delta. This algorithm is based on a type of hypothesis testing (using adaptively collected samples). They prove that this algorithm also has a good regret bound. An ancillary result, and actually the one that I find to be the most exciting result of the paper, is Theorem 4. This shows that for zero-gap problems, UCB1 will essentially balance the number of pulls between the two arms. This is interesting and likely will have neat applications in the future.

Strengths: First, I think the problem is interesting. The authors give plenty of motivation, so I won't repeat it here (it seems realistic to restrict to a finite number of types). I believe the results are correct, though due to time constraints I did not check the proofs in detail. From what I did read, I have confidence in the proofs. In hindsight of the authors' clear explanations, it is quite clear that one should be able to get regret bounds like the ones the authors get in the Delta-agnostic setting (although fully understanding some of the constants is not trivial). Their algorithm is simple, elegant, and how it succeeds in obtaining low regret is quite transparent. I found it a joy to read this paper. As mentioned in the summary, I found Theorem 4 to be especially exciting. This result is definitely of independent interest. I myself have thought about it (I am referring to part i of the result) for a while, without success, in the past. To expand on the authors' result, if we take delta and epsilon to be suitably small constants, then the probability that the pulls between the two arms are badly imbalanced (imbalanced here is modulated by epsilon) decays at a rate close to n^{-3}. This seems quite good. I checked this result in detail and could follow all but one (technical) step, namely, the inequality (*) in the math display between lines 150 and 151 in the supplementary material (an explanation was given in terms of some function rho). I don't doubt the correctness, but simply didn't work this out myself. Theorem 4 (especially part i) should have some interesting applications in the future. The empirical comparison between UCB1 and various versions of Thompson Sampling also is fascinating; I was not aware of how Thompson Sampling can be have so differently for zero-gap problems (although I do now see that there was some literature about this already).

Weaknesses: One weak point is the lower bound, which I feel could be improved at have a 1/(alpha Delta) (though it is unclear if this term also should scale with log n). In fact, the authors current lower bound is weird. If alpha is very low, the lower bound gets worse (i.e., lower)! This lower bound is worse than the standard lower bound for these problems, for any value of alpha (ignoring the constants). Can't you just use a standard lower bound for a two-armed bandit problem? CAB is only harder, right? My score would have been a 9 if the lower bound actually said something more meaningful.

Correctness: I generally believe the results are correct. However, aside from Theorem 4, I did not check all the results in detail. For Theorem 2, I verified the first part of the proof (B.1 in the supplementary material) and am suitably convinced that the overall proof is correct. For Theorem 3, I checked some parts of the proof but I skipped some of the major results (like the ones that give the important constants beta and C_0). It is helpful that the authors have solid reasoning for their results, so if there are some bugs, I imagine they are fixable.

Clarity: The paper is exceptionally clear. Again, it was a joy to read. There are many explanations given to share intuition with the reader.

Relation to Prior Work: I believe the authors are aware of all of the important related work. There is one point that I wonder about, though it would take some work on the part of the authors (but not that much, if they are quite familiar with this paper): I do wonder about the relationship of this work to the results of [20], i.e. "Algorithms for infinitely many-armed bandits". Specifically, it seems that [20] considers a harder situation where the probability that a randomly selected arm is an epsilon-optimal arm is of order epsilon^beta for some constant beta. Now, in your problem, this would instead be some constant probability equal to alpha, where epsilon is selected to be the gap. So, using the results of [20] or a suitably simplified version of their algorithms and analysis, what would one get? It seems like a natural question that deserves at least a couple of sentences.

Reproducibility: Yes

Additional Feedback: Regarding Definition 1, equation (2) in the supplementary material. While (2) seems natural to assume, I do wonder if the authors have examples of algorithms for which (2) provably holds. This would strengthen the analysis.


Review 4

Summary and Contributions: This paper considers a multi-armed bandit problem with countably many arms from a finite set of types (the main focus is on 2 types). The paper explores this setting first under the assumption the optimality gap is known using an explore-then-commit algorithm based on hypothesis testing. This motivates an algorithm that combines hypothesis testing and an optimistic algorithm (e.g. UCB1) for the case the optimality gap is not known. The work provides precise performance guarantees, along with a concentration bound on the fraction each arm is played in UCB1 when all arms have equal mean. The work suggests empirically that the same concentration does not hold for Thompson sampling.

Strengths: - Paper provides new insight on multiple aspects of the MAB problem by studying a novel fundamental problem - The proposed algorithms and guarantees are of substantial interest to the research community - The paper provides a novel result on concentration for UCB type algorithms that does not seems to hold for Thompson Sampling that could be more widely applicable in future works relevant to the community - The paper is very well written and accessible

Weaknesses: - The near-optimal gap-agnostic algorithm for arbitrary finite cardinality T (Appendix G.2) intuitively seems to scale poorly in K as it resets whenever it identifies 2 arms from the same type (meaning another type must be missing), which is more and more likely as K increases. A successive elimination flavored algorithm may be much more efficient. I would like to hear the author's thoughts on this and maybe address briefly in the main text as highlight for potential future direction (clearly out of scope for this paper).

Correctness: I haven't found any issues concerning correctness

Clarity: The exposition is very clear; the main line of argument is easy to follow, and while proofs are provided in the appendix, the main text gives a good intuition about them.

Relation to Prior Work: Relation to prior work is clear

Reproducibility: Yes

Additional Feedback: Overall, I really like this work, it really adds new insight to the large existing body of work on multi-armed bandits and will be highly relevant to the community. POST REBUTTAL: Thanks for the clarifications and comments. While the paper doesn't resolve all open questions, I remain of the opinion that it's a strong accept and would be appreciated by the community. PS I realized this rather late, but it may be worth commenting on the relationship with the heavy-coin problem as well, e.g. Finding a most biased coin with fewest flips (https://arxiv.org/abs/1202.3639) and related works.

[Author Response · NeurIPS 2020]

We greatly appreciate the reviewers' constructive feedback and thank them for their time and interest in our manuscript.

**Clarification on assumptions underlying the model (Reviewer 2).** We consider an infinite population with two
types of arms, each characterized by a *unique* mean reward. We assume that the rewards (not just mean rewards) are
bounded in $[0, 1]$. Algorithm 1 relies on said assumptions alone. However, Algorithm 2 additionally requires knowledge
of the difference between the maximal and minimal elements of the reward *support*. This information informs the
calibration of the $(\theta_m)_{m \geq 1}$ sequence appearing as an input to Algorithm 2. Assumption 1 (line 102, main text) fixes
said difference at 1. Thus, examples of permissible reward distributions include Bernoulli$(0.5)$, Beta$(2, 3)$, Uniform
on $[0, 1]$, etc. However, Uniform on $[0, 0.9]$ is not a permissible reward distribution. We remark that comparisons of
Assumption 1 with conventional assumptions in infinite-armed bandit literature that pertain to regularity-like conditions
in the neighborhood of 1, are not directly relevant. The latter concern the *reservoir distribution* of mean rewards while
Assumption 1 concerns the reward distributions directly. We intend to make this distinction clearer in the revision.

**Remarks on lower bound and phase transition phenomenon (Reviewers 2,3).** The multiplicative factor of $\alpha(1-\alpha)$
is indeed a proof artifact and can be improved upon. Under a mild assumption on $\alpha$, we have been able to eliminate this
factor altogether, resulting in an asymptotic scaling that is on par with the finite-armed problem. Indeed, this reconciles
with point (i) (line 275, main text). The modified proof assumes $\alpha \in (0, 1)$ fixed and invariant w.r.t. the horizon, and is
not amenable to the boundary cases of $\alpha \in \{0, 1\}$. Thus, quite remarkably, a phase transition occurs in the lower bound
from linear to logarithmic at $\alpha = 0$ while another from logarithmic to 0 occurs at $\alpha = 1$. We appreciate the reviewers'
feedback on this aspect and intend to incorporate the necessary corrections in our revision. The intent behind restriction
to consistent policies is to facilitate a direct comparison with the classical Lai and Robbins proof technique for the
finite-armed bandit problem. Indeed, the assumption of asymptotic consistency is restrictive, but more generic policy
classes are unwieldy for lower bound proofs due to reasons stemming from the combinatorial nature of our problem.

**Clarifications on Algorithm 1 (Reviewers 1,2).** The algorithm proceeds by "selecting" two arms at random and
"pulling" each arm $m$ times. If the separation between the empirical mean rewards is large enough, the algorithm
commits to the empirically better arm; else discards "arm 2," randomly selects a new arm in its place, labels it "2" and
pulls it $m$ times. The process is repeated thereafter. Throughout the algorithm's lifetime, arm 1 stays fixed while the
label "2" potentially recirculates between different arms. In the upper bound of Theorem 2, the $o(1)$ term is independent
of $(\alpha, \Delta)$ and can be bounded above by a true absolute constant (no dependence on free parameters of the algorithm).

**Clarifications on Algorithm 2 (Reviewers 1,2).** The algorithm expends all of its sampling effort on a given hetero-
geneous consideration set of arms only *in expectation*, not with probability 1. In fact, the probability of discarding a
heterogeneous consideration set and reinitializing the algorithm is bounded away from 0 at all times. This leads to a
multiplicatively larger regret with $\beta^{-1}$ as the inflation factor, as opposed to merely an additive loss. $\beta$ is a lower bound
on the probability of never discarding a heterogeneous consideration set and depends on the reward distributions alone.
In short, the regret is inflated by $\beta^{-1}$ due to exploration of *new* arms happening throughout the algorithm's lifetime.

**Remarks on $\beta$ appearing in the upper bound of Theorem 3 (Reviewer 2).** The behavior of $\beta$ vis-à-vis $\Delta$ is hard
to characterize mathematically. However, we empirically observe that $\beta$ scales with $\Delta$ linearly on "well-separated"
instances. While absent presently, we intend to include this observation in the revision. The implication is that the
regret scales as $1/\Delta^2$ on well-separated instances, as opposed to the classical $1/\Delta$ scaling achievable in finite-armed
bandits. In the small $\Delta$ regime, however, the precise characterization of the rate at which $\beta$ vanishes is mathematically
challenging and remains an open problem. Although this precludes quantification of the minimax regret and thus
a comparison to the infinite-armed problem with a rich (infinite) set of types (a well-studied problem), our paper is
focused on instance-dependent bounds alone. We intend to include a relevant discussion on this matter in the revision.

**Improvements to Algorithm 2 (Reviewers 2,4).** The reviewers are referred to points (iii) and (iv) in the "Miscellaneous
remarks" section (line 275, main text) concerning potential improvements to Algorithm 2. Similar improvements to the
extension of Algorithm 2 to $K$ types are also possible. We reemphasize that the consideration set size must at all times
be fixed at $K$, in a $K$-typed setting. Any more is redundant while a reduction in size shall cause the algorithm to spend
a positive fraction of its sampling effort (in expectation) on inferior consideration sets, thereby incurring linear regret.

**Performance of existing algorithms for infinite-armed bandits (Reviewers 2,3).** A suitable modification of the
algorithm in [20] (references) provably achieves poly-log regret on our problem, a significant performance degradation.

**UCB vs. Thompson Sampling on the zero gap problem.** In a nutshell, UCB's faster convergence on the zero gap
problem has to do with the presence of the $\sqrt{\log n}$ additive bias in the UCB score. Thompson Sampling lacks in
this regard, thereby causing the fraction of samples from a given arm to converge to a non-degenerate limit, evident
empirically. This is also suggestive of UCB being better suited to adversarial settings than Thompson Sampling.
However, this is only an empirical observation and remains a conjecture at the moment. More work on the matter is
presently underway. While we have not been able to address here each and every remark by the reviewers, we have
taken due note of all the points and hope that the most substantive ones have been satisfactorily answered in this rebuttal.

[Meta-Review · NeurIPS 2020]

The reviewers all liked the paper; beside the new model (and its analysis) they particularly liked "anti-concentration" result for UCB. Nevertheless, they pointed out a few issues which should be addressed in the final version. The most important ones are (i) the issue with beta should be properly discussed; (ii) it would be good to have a motivating example which actually matches the model (unlike the current ones); (iii) the algorithm for more than 2 arms seems quite wasteful.